# Cloud impacts on photochemistry: building a climatology of photolysis rates from the Atmospheric Tomography mission

Samuel R. Hall[1], Kirk Ullmann[1], Michael J. Prather[2], Clare M. Flynn[2], Lee T. Murray[3], Arlene M. Fiore[4,5], Gustavo Correa[4], Sarah A. Strode[6,7], Stephen D. Steenrod[6,7], Jean-Francois Lamarque[1], Jonathan Guth[8], Béatrice Josse[8], Johannes Flemming[9], Vincent Huijnen[10], N. Luke Abraham[11,12], Alex T. Archibald[11,12]

[1] Atmospheric Chemistry Observations and Modeling Laboratory, National Center for Atmospheric Research, Boulder, CO 80301, USA
[2] Department of Earth System Science, University of California, Irvine, CA, USA
[3] Department of Earth and Environmental Sciences, University of Rochester, Rochester, NY, USA
[4] Department of Earth and Environmental Sciences, Columbia University, NY, NY, USA
[5] Lamont-Doherty Earth Observatory of Columbia University, Palisades, NY, USA
[6] NASA Goddard Space Flight Center, Greenbelt, MD, USA
[7] Universities Space Research Association (USRA), GESTAR, Columbia, MD, USA
[8] Centre National de Recherches Météorologiques, CNRS-Météo-France, UMR 3589, Toulouse, France
[9] European Centre for Medium-Range Weather Forecasts, Reading, UK
[10] Royal Netherlands Meteorological Institute, De Bilt, the Netherlands
[11] Department of Chemistry, University of Cambridge, Cambridge, U.K.
[12] National Centre for Atmospheric Science, U.K.

*Correspondence to*: Michael J. Prather (mprather@uci.edu)

**Abstract.** Measurements from actinic flux spectroradiometers on board the NASA DC-8 during the Atmospheric Tomography (ATom) mission provide an extensive set of statistics on how clouds alter photolysis rates (J-values) throughout the remote Pacific and Atlantic Ocean basins. J-values control tropospheric ozone and methane abundances, and thus clouds have been included for more than three decades in tropospheric chemistry modeling. ATom made four profiling circumnavigations of the troposphere capturing each of the seasons during 2016-2018. This work examines J-values from the Pacific Ocean flights of the first deployment, but publishes the complete Atom-1 data set (29 July to 23 August 2016). We compare the observed J-values (every 3 sec along flight track) with those calculated by nine global chemistry–climate/transport models (globally gridded, hourly, for a mid-August day). To compare these disparate data sets, we build a commensurate statistical picture of the impact of clouds on J-values using the ratio of J-cloudy (standard, sometimes cloudy conditions) to J-clear (artificially cleared of clouds). The range of modeled cloud effects is inconsistently large but fall into two distinct classes: (1) models with large cloud effects showing mostly enhanced J-values aloft and or diminished at the surface; and (2) models with small effects having nearly clear-sky J-values much of the time. The ATom-1 measurements generally favour large cloud effects but are not precise or robust enough to point out the best cloud-modeling approach. The models here have resolutions of 50-200 km and thus reduce the occurrence of clear sky when averaging over grid cells. In situ measurements also average scattered sunlight over a mixed cloud field, but only out to scales of 10s of km. A primary uncertainty remains in the role of clouds in chemistry, in particular, how models average over cloud fields, and how such averages can simulate measurements.

**1 Introduction**

Clouds visibly redistribute sunlight within the atmosphere, thus altering the photolytic rates that drive atmospheric chemistry (J-values), as well as the photosynthesis rates on the land and in the ocean. These J-values drive the destruction of air pollutants and short-lived greenhouse gases. The NASA Atmospheric Tomography Mission (ATom, 2017; Wofsy et al., 2018), in its

charge to measure the chemical reactivity over the remote ocean basins, has measured J-values while profiling the troposphere (0 – 12 km). These measurements reveal a statistical pattern of J-values over different geographic, altitude and cloud regimes, which directly challenges current atmospheric chemistry models and provides a new standard test of cloud effects. The observations quantify how clouds alter photochemistry and are compared here with parallel analyses from nine global atmospheric chemistry models.

Since the early models of atmospheric chemistry, the scientific community has tried various approximations and fixes for "those pesky clouds". Overhead clouds can shadow the sun, resulting in diminished J-values beneath and within the lower parts of thick clouds. Cloud scattering results in enhanced J-values above and within the tops of clouds. For ideal clouds – uniform layers from horizon to horizon – the models have developed a variety of methods to approximate the 1D radiative

transfer and calculate the J-values relative to a clear sky (Logan et al., 1981; Chang et al., 1987; Madronich, 1987; Wild et al., 2000; Lefer et al., 2003; Williams et al., 2006; Palancar et al., 2011; Ryu et al., 2017). More realistic treatment of clouds is important in global chemistry as well as air pollution (Kim et al., 2015). For the most part, these chemistry models are provided with the cloud properties and fractional coverage for each grid cell in a column, make some assumptions about the overlap of cloud layers, and then solve the 1D plane-parallel radiative transfer equation at varying levels of accuracy. In a 3D

world, however, adjacent clouds can block the sun or scatter light even when there are clear skies overhead. Also in a 3D world, a sunlit adjacent cloud that is not overhead can increase J-values. It is nigh impossible to specify the 3D cloud fields at ~1 km scale along the ATom flight paths or any similar mission: 3D high-resolution cloud fields on this are available, but these are limited to CloudSat-CALIPSO afternoon overpasses extended to 200-km wide swaths (see Barker et al., 2011; Miller et al., 2014). Anyway, none of the standard global models can deal with such a 3D radiative transfer problem. So accepting

the model limitations and the inability to match individual measurements, we use the statistics of observed J-values, enhanced or diminished relative to a clear sky, and ask if the models' many approximations for the radiative transfer in cloud fields can yield those same net results.

This paper presents the statistical distribution of the measured J-values using the CAFS instrument (Charged-coupled device

Actinic Flux Spectroradiometers: Shetter and Mueller, 1999; Petropavlovskikh et al., 2007) from the first ATom deployment (29 July through 23 August 2016). We build up statistics of the observed J-value relative to that calculated for a clear sky under similar conditions. The chemical reactivity of the troposphere (see Prather et al., 2017) is generally proportional to these

changes in J-values, and thus modeling of the variability of clouds is critical for modeling the lifetime of $CH_4$ and the cycling of tropospheric $O_3$.

So what makes this model versus measurement comparison different? The atmospheric chemistry community has a history of such comparisons, including photolysis rates, dating to the early ozone depletion assessments (Nack and Green, 1974; M&M, 1993) and continuing to recent multi-model projects (Olson et al., 1997; Crawford et al. 2003; PhotoComp, 2010). These comparisons have been limited mostly to simplistic atmospheric conditions and measurements made under clear skies or with uniform 1D cloud layers for which an accurate solution can usually be calculated. We introduce here the ability to use CAFS all-sky measurements made under a semi-objective sampling strategy (i.e., ATom's tomographic profiling makes pre-planned slices through the troposphere, limited by available airports, diverting only for dangerous weather). We test the collective treatment of clouds and radiation in models insofar as they can match the observed statistics of J-values. This approach gets to the core of atmospheric photochemistry by combining the range of assumptions and parameterizations for clouds in the models including, among others, cloud optical depths and scattering phase functions, two-stream or multi-stream radiative transfer, cloud overlap, or even just parametric correction factors.

Section 2 describes the measurements and models, including how the observational statistics are compiled, what protocol the models used, and how they included cloud fields. This section also documents the differences in mean J-values, which can be as large as 30% in either cloudy or clear conditions. Section 3 introduces the statistical distributions based on the ratio of full-sky including clouds to clear-sky. Section 4 examines modeling errors and improvements related to this comparison. In the concluding Section 5, we discuss the current range in modeling cloud effects and how new observational constraints can be developed and used to build better models.

## 2 Measuring and Modeling J-values under Realistic Cloudy Skies

Here, we focus on two J-values: $O_3 + h\nu \rightarrow O_2 + O(^1D)$ designated J-O1D; and $NO_2 + h\nu \rightarrow NO + O(^3P)$ designated J-NO2. These J-values are the most important in driving the reactive chemistry of the lower atmosphere, and each emphasizes a different wavelength region with response to atmospheric conditions. J-O1D is driven by short wavelengths (<320 nm), where $O_3$ absorption and Rayleigh scattering control the radiation; whereas J-NO2 is generated by longer wavelengths, where $O_3$ absorption is not important and Rayleigh scattering is 2 times smaller. Thus, J-NO2 is the more sensitive J to clouds. Both CAFS and the models accept the same spectral data – cross sections and quantum yields from recent assessments (Atkinson et al., 2004; Burkholder et al., 2015) – but the implementation (solar spectrum, Rayleigh scattering, wavelength integration, temperature interpolation) may be different.

Spectrally resolved CAFS measurements of actinic flux (280-650 nm) are used to calculate in situ J-O1D and J-NO2. These observed J-values are all-sky J-values and include incidences when the sky is effectively clear of clouds. We designate these all-sky J-values as J-cloudy in both measurements and models to contrast them with the artificially cloud-cleared J-values denoted J-clear. For J-clear, CAFS uses the Tropospheric Ultraviolet and Visible (TUV) radiative transfer model (Madronich and Flocke, 1999). The model is run with an eight-stream discrete ordinate radiative transfer method with a pseudo-spherical modification to generate actinic fluxes with a 1-nm wavelength grid from 292-700 nm. The calculation is run with no clouds and no aerosols, a fixed surface albedo of 0.06, and applies ozone columns from the satellite Ozone Monitoring Instrument (Levelt et al., 2006; Veefkind et al., 2006). CAFS and TUV spectra are processed using the same photolysis frequency code to ensure that the same quantum yield, absorption cross section, and temperature and pressure dependence relationships are applied to the measured and modeled spectra. The strong connection between measurement and model has been established in past campaigns (Shetter et al., 2002, Shetter et al., 2003, Hofzumahaus et al., 2004). The ATom CAFS data set used here is a value-added product beyond the standard ATom output (Wofsy et al., 2018), and it is archived with this paper. Only data from the first deployment, ATom-1, were available when this paper was being prepared.

Global chemistry models cannot be used productively in comparisons with individual CAFS observations as noted above, but a statistical comparison of the ratio J-cloudy to J-clear is a useful climatological test. It is difficult for the models to simulate CAFS global data unless there is a very careful sampling strategy to match albedos from the ATom flights over land and cryosphere. Thus, we focus on the 2 oceanic blocks in the Pacific for which we have a large number of measurements with high sun in ATom-1. See also discussion of ocean surface albedo variations in Section 4. The CAFS statistics are derived from the ATom-1 deployment and selected for 2 remote geographic blocks in order to compare with the models: (block 1) Tropical Pacific, 20S-20N x 160E-240E; and (block 2) North Pacific, 20N-50N x 170E-225E.

The models here include the 6 original ones used in the ATom reactivity studies (Prather et al., 2017) plus 3 additional European global chemistry models. They are described more fully in Table 1, and are briefly designated as: GEOS-Chem (GC); GFDL AM3 (GFDL); GISS Model 2E1 (GISS); GSFC GMI (GMI); ECMWF IFS (IFS); MOCAGE (MOCA); CESM (NCAR); UCI CTM (UCI); and UM-UKCA (UKCA). Additional model information and contacts are given in Supplementary Table S1. All models submitted global 4D fields (latitude by longitude by pressure for 24 hours) for one day in mid-August using their standard treatment of clouds (designated all-sky or "cloudy" here) and then a parallel simulation with clouds and aerosols removed (designated clear-sky or "clear'). In addition to its correlated cloud-overlap model with multiple quadrature column atmospheres to calculate an average J-value, UCI also contributed a model version using the B-averaging of cloud fractions (Briegleb, 1992) used by most models, designated UCIb (see Prather, 2015). Several models ran the clear-sky case without clouds but with their background aerosols. Globally, aerosols have a notable impact on photolysis and chemistry (Bian et al., 2003; Martin et al., 2003), but over the middle of the Pacific Ocean (this analysis), the UCI model with and without aerosols shows mean differences of order ±½%.

Modeling the effect of clouds on J-values began with early tropospheric chemistry modeling. One approach was to perform a more accurate calculation of generic cloud layers offline and then apply correction factors to the clear-sky J's computed inline: e.g., an increase above the cloud deck and a decrease below (Chang et al., 1987). Another approach used a climatology of

overlapping cloud decks to define a set of opaque, fully reflecting surfaces at different levels: e.g., the J's would be averaged over these sub-grid column atmospheres (Logan et al., 1981; Spivakovsky et al., 2000). As 3D tropospheric chemistry models appeared, the need for computationally efficient J-value codes led to some models ignoring clouds and others estimating cloud layers and applying correction factors to clear-sky J's. With the release of Fast-J (Wild et al., 2000), some 3D models started using a J-value code that directly simulated cloud and aerosol scattering properties with few approximations. The next

complexity, based on general circulation modeling, included fractional cloud cover within a grid cell and thus partial overlap of clouds in each column (Morcrette and Fouquart, 1986; Briegleb, 1992; Hogan and Illingworth, 2000). This approach later moved on to chemistry models (Feng et al, 2004; Liu et al., 2006; Neu et al., 2007). Monte Carlo solutions for the numerous independent column atmospheres generated by cloud overlap were developed for solar heating (MCICA: Pincus et al., 2003), but random, irreproducible noise was not acceptable in deterministic chemistry-transport models. The Cloud-J approach

(Prather, 2015) developed a scale-independent 1D method for cloud overlap based on vertical decorrelation lengths (Barker, 2008a, 2008b). The chemistry models here use a variety of these methods, which range from lookup tables with correction factors, to Fast-J single column, to cloud overlap treatments with Cloud-J, see Table 1. Currently these models do not attempt to define 3D cloud structures within a grid square, the approach needed to match individual CAFS J's.

The CAFS data were collected from ATom-1 during its 10 research flights from 29 Jul to 23 Aug 2016. It was not possible for the models simulate each flight path, including clouds and local solar zenith angles (SZA) for each measurement. Not all of the models could run with 2016 meteorology, and thus we asked for a day in mid-August and treat that (rightly or wrongly) as typical of the cloud statistics during ATom-1. The meteorological years are listed in Table 1. This simplification made it possible to attract large participation, but of course it limits the ability to claim that the model statistics are a robust climatology.

ATom flights are mostly in daylight and hence a large proportion of CAFS measurements occur at high sun, cos(SZA) > 0.6, with more than half at cos(SZA) > 0.8 (Supplementary Figure S1). The models report hourly J-O1D and J-NO2 globally over 24 hours, and thus all have a similar distribution of cos(SZA) but with a greater proportion at cos(SZA) < 0.4 than the CAFS data. We restrict these comparisons to high-sun, cos(SZA) > 0.8, to reduce three-dimensional effects that are not modeled here, which leaves 11,504 (block 1) and 4,867 (block 2) measurements for the CAFS/TUV 3-sec averages. For the models,

the number of hourly samples with cos(SZA) > 0.8 is about 18% for both blocks and thus the number of samples depends on model resolution (e.g., 240,000 for UCI and 1,400,000 for NCAR in block 1). Although the analysis here is limited to the Pacific blocks, the global model data are archived with this paper.

A quick look at J-cloudy (all sky) profiles of J-O1D and J-NO2 for CAFS (Figure S2) shows a basic pattern also seen in models. Both J's are larger in the upper troposphere where the direct sunlight is more intense; but in the northern Pacific, larger cloud cover and more scattered light almost reverses this pattern with enhanced J's at lower altitudes (>600 hPa). Comparing the variances of J-cloudy (CAFS) and J-clear (TUV) for the tropical Pacific, the J-NO2 variability is driven almost

entirely by clouds as expected, while the J-O1D variability is driven firstly by $O_3$ column and sun angle (both CAFS and TUV), while there are clearly cloud contributions (CAFS only) at lower altitudes (>700 hPa).

Figure 1 shows a full comparison of the CAFS J profiles with the 10 model results in 4 panels (2 J's x 2 geographic blocks). The CAFS J's fit within the range of models; their shape is matched by most models; but the model spread of order 20-30% is

hardly encouraging. Differences in these average profiles can have many causes: temperature and $O_3$ profiles; spectral data for both J-O1D and J-NO2; ways of integrating over wavelength; surface albedo conditions; treatment of Rayleigh scattering; basic radiative transfer methods; SZA; and, of course, clouds. In typical comparisons we try to control for these differences by specifying as many conditions as possible; but here we want to compare the 'natural' J's used in their full-scale simulations (e.g., Lamarque et al., 2013) and thus leave each model to its native atmospheres, spectral data, algorithms and approximations.

The models show a much tighter match in J profiles under clear-sky conditions (Figure 2). Typically, 8 of the models fall within 10% of their collective mean profile. Some models are obviously different in J-clear (GISS and MOCA for J-O1D, MOCA for J-NO2), and these differences carry through to J-cloudy (Figure 1). A most important factor in J-O1D is the $O_3$ column, and Figure S3 shows the modeled $O_3$ columns for August compared with 8 years of OMI observations. MOCA,

NCAR and IFS have low tropical $O_3$ columns, <250 DU vs. observed ~265 DU, which could lead to higher J-O1D, but this effect is seen only in MOCA. UKCA has higher tropical columns, >300 DU, which might explain why their J-O1D lies in the lower range of the models. Some results, like MOCA's J-NO2 and GISS's J-O1D point to differences in the implementation of spectral data (e.g., wavelength integration, solar spectrum, temperature interpolation). We expect GC and GMI to be alike since they both use MERRA-2 cloud fields and Fast-J with Briegleb averaging: indeed, they match well except for J-O1D in

the tropical Pacific, yet, they report similar tropical ozone columns.

The ratio of J-cloudy to J-clear, shown in Figure 3, cancels out many of the model differences in Figures 1 and 2, and as expected GC and GMI are nearly identical. The cloudy:clear ratio identifies new patterns in model differences whereby some models have ratios close to 1 throughout the troposphere (especially in the tropics) while others have ratios >1.1 at altitudes

above 800 hPa and <0.9 below 900 hPa. In a recent model intercomparison with specified chemical abundances (Prather et al., 2018), we found that the tropospheric photochemistry of $O_3$ and $CH_4$ responded almost linearly to cloudy-clear changes in J-values (see Figure S4, data not shown in Prather et al., 2018). Thus the differing impact of clouds on J-values seen in Figure 3 will have a correspondingly impact on global tropospheric chemistry (see Liu et al., 2009).

## 3 The Statistical Distribution of J-cloudy to J-clear

The average ratio of J-cloudy to J-clear (Figure 3) provides only a single measure of the impact of clouds. The CAFS data provide a more acute measure by sampling the range of cloud effects (enhance or diminish J-values) and their frequency of occurrence. A quick look at this range in CAFS data is shown in Figure S5 with the probability of occurrence of the cloudy-to-clear ratio defined as ln(J-cloudy/J-clear) and designated rlnJ. Each curve is normalized to unit area with the Y-axis being probability per 0.01 bin (~ 1 %) in rlnJ.

We expect that enhancements in J's occur above clouds and diminishments below, and this is borne out in Figure S5. In the marine boundary layer (900 hPa – surface), there are a greater number of rlnJ < -0.10 with fewer rlnJ > 0.00. In the mid-troposphere layer (300 – 900 hPa), there are frequent occurrences of rlnJ > +0.10 particularly in the North Pacific where lower level clouds are more extensive than in the tropics. Likewise, there are times when rlnJ is < 0.00 in the mid-troposphere when clouds lie overhead. In upper tropospheric layers (100 – 300 hPa), most of the optically thick clouds are below and rlnJ > 0.00 is dominant. Thus, our analysis here breaks the atmosphere into these 3 layers. All figures in this section will be displayed as 2-block by 3-layer panels with part a (J-O1D) and part b (J-NO2).

### 3.1 Modeling the distribution of J-values

The probability distribution of rlnJ for J-O1D (Figure 4a) and J-NO2 (Figure 4b) shows highly varied patterns across the models, but with some consistency. The models and CAFS are not exactly a match, but again, there are some encouraging patterns. The peak rlnJ distributions for CAFS will be broadened because the real variation in ocean surface albedo is not simulated in TUV (see discussion in Section 4), but this is expected to be of order ±2%, and so the overall width (10 to 20 %) reflects cloud variability. The modeled and measured distributions are asymmetric and skewed toward rlnJ > 0 in the free troposphere (100 – 900 hPa), and toward rlnJ < 0 in the boundary layer. This pattern is expected since J's are enhanced above clouds (100 – 900 hPa) and diminished below. There will also be some occurrences of rlnJ < 0 in the free troposphere when thick clouds are overhead, but none of the models come close to the CAFS frequency of these occurrences. In general and as expected, this brightening above and dimming below is more evident for J-NO2 than for J-O1D. Another feature that is somewhat consistent across models and observations: the wings of the rlnJ distribution are wider in the North Pacific than the Tropics. In the boundary layer, observations and models show the reverse with greater cloud effects (dimming, rlnJ < 0) in the Tropics. Although all the models show this shift from rlnJ > 0 to rlnJ < 0 in the peak of their distributions, only a few (GISS, MOCA, NCAR, UCI) have broad wings of large cloud shielding (rlnJ < -0.1). These models calculate such broad wings for both Tropics and North Pacific; whereas CAFS only shows this in the Tropics.

These diagnostics also identify some CAFS anomalies that have no physical basis in the current models. For example, low-level (surface to 900 hPa) observed cloud enhancements (rlnJ > 0.025) observed in the tropical Pacific (particularly J-O1D)

does not appear in the models. Similarly cloud diminishments in the upper troposphere do not appear with the modeled cirrus. A more thorough analysis of the CAFS rlnJ with the added deployments should examine if these differences are due to 3D radiative transfer effects, ocean albedo variations, or missing cloud types in the models. Perhaps, we will identify sun-cloud geometry conditions simply not possible in the 1D cloud models.

Figures 5a (J-O1D) and 5b (J-NO2) provide another view of the rlnJ statistics. Figure 5 has the same 6 blocks as figure 4, but plots 5 quantities schematically in a single line describing each model's statistics: percent occurrence of diminishment (rlnJ < –0.025), average rlnJ diminishment, percent occurrence of nearly clear sky, percent occurrence of enhancement (rlnJ > +0.025), and average rlnJ enhancement. The horizontal line has total length of 100% with the thin line (clear sky) centered on 0, see

the extended legend in Figure 5b.

Overall, four models (GC, GFDL, GMI, UKCA) have unusually narrow peak distributions of rlnJ ~ 0, indicating lesser cloud effects on the J's. The other 6 models (GISS, IFS, MOCA, NCAR, UCI, UCIb) show a much greater range in J's, with a larger fraction perturbed by clouds (enhanced or diminished by more than 10%). The CAFS observations generally support this latter

group. There are individual model anomalies that may point to unusual features: MOCA alone has a peak frequency of enhanced J's at rlnJ ~ 0.05 in the free troposphere; three models (NCAR, UCI, UCIb) show the largest extended frequency of |rlnJ| > 0.10 in the middle troposphere; UKCA is consistently the most "clear sky" model. These model differences are not simply related to the model cloud fields, see discussion of Figure S6 below.

Immediately above and below extensive thick cloud decks the dimming/brightening of J's exceeds the plotted range of rlnJ of ±0.3 (a factor of 1.35). Most such cloud decks occur around 900 hPa and so the largest brightening occurs in the 100-900 hPa levels, and the greatest dimming at >900 hPa. The top two rows in Figure 4 give the fraction of samples for which rlnJ > 0.3 on the right side of each plot; the bottom row gives, on the left side, the fraction for which rlnJ < -0.3. The categorization of models and measurements is not simple as many models have shifting magnitudes of this large-scale brightening or dimming.

A few models consistently lack these large changes in J's (GC, GMI), and a few always have them (NCAR, UCIb). Large CAFS values are clearly evident in both J's for only 2 of the 6 cases: >900 hPa in Tropical Pacific (13 – 15 % of all J's) and 300-900 hPa in North Pacific (8 – 15 %). For these cases the CAFS extreme fractions are consistent with at least 4 of the models. Any possible CAFS bias in rlnJ due to TUV modeling (±0.05) is unlikely to affect these results. These extreme fractions, however, are likely sensitive to any sampling bias of flight path with respect to thick cloud decks, and this needs to

be assessed with model sampling that matches the ATom-1 profiles of that period.

The large geographic blocks were chosen to acquire representative sampling from the models that would be repeatable over time. It was not possible to acquire month-long or multi-year diagnostics from all the models, and so with the available model results (24 hours from a day in mid-August) we sub-sample the broad tropical Pacific block (20S-20N x 160E-240E) into a

west (160E-200E), east (200E-240E) and dateline (175E-185E) blocks. The results of these sub-sampled statistics are shown for J-NO2 in Figures S7a (100-300 hPa) and 7b (surface-900 hPa) using the same format as Figure 5. Each of the 18 panels in Figure 7ab represents a single model and a single pressure level for which there are 7 bars (sampled distributions of rlnJ). The top 4 bars show the full tropical Pacific (as in Figure 5b) and then the 3 sub-sampled regions. The next 3 bars are the previously sampled statistics (north Pacific, global 50S-50N, and tropical Pacific CAFS). The CAFS bars are the same in each 9-panel plot. Six models show no difference across the sub-sampling (NCAR, GFDL, IFS, UCI, UKCA, MOCA), while three models using MERRA cloud fields in one way or another (GC, GMI, GISS) show weaker cloud effects in the east half of the region. For the most part, the sub-sampled regions have similar statistics that are distinct from the CAFS observations. Thus our model statistics over a large block may be a representative climatology of cloud effects on J-values. Further sampling tests are recommended for follow-on work, especially to determine if the distinct east-west differences for the 3 MERRA-cloud models are a standard feature.

## 3.2 Analyzing cloud effects

With a graphical synopsis of the rlnJ probability distributions in Figures 5ab, some model features become more obvious. We define nearly cloud-free conditions as being within ±2.5 % of clear-sky J's, and show the frequency of these with the length of the thin line in the center of the plots. Starting with the J-O1D in the upper Tropical Pacific, we find 5 models (Group 1: GC, GFDL, GMI, GISS, UKCA) show no effect of clouds more than 50 % of the time. The other 40 – 50 % of the time, they show enhanced J-O1D, cloud brightening expected from clouds below (thick lines on the right side of the plot). For the other 5 models (Group 2: IFS, MOCA, NCAR, UCI, UCIb), these clear-sky equivalent J's occur only 10-20% of the time, with cloud brightening enhancements occurring at 80 – 90 %. Surprisingly, both model groups show the same average magnitude, +10 % (X's on the right side), for their enhanced J's. Thus Group 2 models will have systematically greater J-O1D in the upper Tropical Pacific than the Group 1 models (i.e., the 10 % enhancement occurs twice as often). In the North Pacific, this pattern holds although both groups show slightly greater frequency of enhanced J's, e.g., 50 to 60 % for Group 1 and 80 to 90 % for Group 2. For J-NO2, the results are similar, but with greater average magnitude of enhancement for cloudy skies (20 % vs. 10 %) and a slightly greater frequency of occurrence (thick line on the right). For this upper tropospheric layer, none of the models show significant occurrence of diminished J-values from overhead clouds (rlnJ < -0.025) as seen in CAFS for 2 – 12 % of the measurements.

In the middle troposphere (300 – 900 hPa, middle panels), the patterns in clear-sky frequency remain unchanged, but there is a shift to cloud dimming for 5 to 20% of the time. This shift to more cloud obscuration is much greater in CAFS than in any model. Group 1 models show consistently more frequent cloud obscuration (10-20%) than do Group 2 models (5-10%). When cloud brightening occurs (both CAFS and models), the magnitude of enhancement is greater than in the upper troposphere. Such a pattern is consistent with the simple physics that J's are greater immediately above a cloud than high above it.

In the boundary layer, most clouds are above, and cloud obscuration leads to increased occurrence of rlnJ < -0.025 compared to the middle troposphere (in both CAFS and models). Even though the frequency changes, the average magnitude of diminishment when there is cloud obscuration (denoted by X's) does not change much across models in either region. For CAFS in the Tropical Pacific, however, the diminishment when there is cloud obscuration is much larger in the boundary

layer. The modeled shifts in frequency of occurrence from enhanced to reduced J's are dramatic, but still the Group 1 pattern of nearly 50% clear-sky J's persists. This results in Group 2 having a much larger frequency of diminished J's (60-80%) as compared with Group 1 (20 – 40 %).

Using CAFS data to define nearly cloud-free conditions is imperfect. Potential biases exist with TUV modeling of J-clear and

are related to albedo as discussed in Section 4. In addition, the CAFS data does not represent a true climatology due to flight planning and flight operations that tend to avoid strong convective features and thick cloud decks, particularly near the surface. Such biases can shift the distribution as well as widen it through noise, and this may explain some of the increased width of the CAFS peak and the 1 to 2% offsets of the clear-sky peaks in Figures 4. It is difficult to select between Group 1 and 2 using CAFS. The CAFS clear-sky fraction lies between that of the two groups in the upper troposphere but becomes narrower in

the boundary layers, more closely matching that of Group 2. Given that a number of processes can lead to broadening of the CAFS distribution, it is likely the sharps peaks in Figure 4 (and wide central lines in Figure 5) of Group 1 are unrealistic.

These model differences have no obvious, single cause. The modeled profiles of cloud optical depth (COD) and cloud fraction (CF) for both geographic blocks are shown in Figure S6 (note the logarithmic scale for COD). The total COD is given (color-

coded) in each block. The profiles show very large variability that is hard to understand. For example, GFDL and GISS show the largest COD, yet both are in Group 1 with the largest fraction of clear sky. Overall the total COD does not obviously correlate with the two groups. Likewise, CF is not a predictor for the Group. It is likely that model differences are driven by the treatment of fractional cloud cover. For example, GMI (Group 1) and UCI (Group 2) have very similar cloud optical depths (COD) and cloud fractions (CF) in the lower troposphere as shown in Figure S6. They also use similar J-value codes

including spectral and scattering data based on the Fast-J module. Yet, they have a factor of 2 difference in the frequency of nearly cloud-free sky as shown in Figure 5. Compared to GMI, UCI shows an overall greater impact of clouds with 2x larger frequency of cloud brightening in the upper troposphere and 2x larger occurrence of cloud dimming in the boundary layer. These differences could be caused by GMI calculating J-values with a single column atmosphere (SCA) containing clouds with Briegleb ($CF^{3/2}$) averaging and UCI calculating J-values with four quadrature column atmospheres (QCAs), see Table 1.

Unfortunately, when UCI mimics the B-averaging (with model UCIb), the differences remain. See further discussion of Figure S6 in Section 4.1.

## 4 Model Difficulties and Development

The J-value statistics here depend on (i) the cloud fields used in the models, (ii) the treatment of cloud overlap statistics, (iii) the radiative transfer methods used, and (iv) the spectral data on sunlight and molecular cross sections. These components are deeply interwoven in each model, and it is nearly impossible to have the models adopt different components except for (iv), where there has been a long-standing effort at standardization (e.g., the regular IUPAC and JPL reviews of chemical kinetics, Atkinson et al., 2004; Burkholder et al., 2015). These components are briefly noted in Table 1.

### 4.1 Cloud optical depths and overlap statistics

The models reported their average in-cell cloud optical depth (per 100 hPa) and cloud fraction over the two Pacific blocks in Figure S6. Averaged cloud optical depths (defined for the visible region 500-600 nm) all tend to peak below 850 hPa in the tropics and decline with altitude. There is clear evidence of mid-level (400-800 hPa) clouds, but only small COD (total $< 0.25$) at cruise altitudes (100-300 hPa). The North Pacific block has 2-4x larger low-altitude COD. The plotted cloud fraction (CF) is the COD-weighted average over 24 hours and all grid cells in the block. Note that for COD the cloud is spread over each model layer, and hence the in-cloud optical depth is estimated by COD/CF. CF is high, 5-15% below 850 hPa, drops off with altitude as does COD, but peaks at 10-20% near 200 hPa corresponding to large-scale cirrus. Some of these differences in COD and CF are large enough to explain model differences; but there is no clear pattern between J-values and clouds as noted in Section 3.2. A more thorough analysis and comparison of the modeled cloud structures would involve the full climate models and satellite data (Li et al., 2015; Tsushima et al. 2017; Williams and Bodas-Salcedo, 2017), beyond the scope here.

### 4.2 Sensitivity of rlnJ to small cloud optical depth

To relate total COD to a shift in rlnJ, the UCI offline photolysis module Cloud-J was run for marine stratus (CF = 1) with a range of total CODs from 0.01 to 100. The cloud was located at about 900 hPa and rlnJ evaluated at 300 hPa. The plot of rlnJ vs log COD for a range of SZA is shown in Figure S8. A 10 % enhancement (rlnJ = +0.10) occurs at COD = 5 for J-O1D and COD = 3 for J-NO2, demonstrating the greater sensitivity of J-NO2 to clouds. Thus model average total COD (ranging from 0.8 to 11 in Figure S6, assuming CF=1) should produce large shifts in rlnJ. Marine stratus with typical COD ~10 or more would produce rlnJ-O1D of +0.16 and rlnJ-NO2 of +0.30. Thus clear-sky J-values (defined here as ±0.025 in rlnJ) require COD $< 1$ for J-O1D and $< 0.3$ for J-NO2. A COD ~ 1 is not that large since these clouds are highly forward scattering and have an isotropic-equivalent optical depth that is 5x smaller.

### 4.3 Averaging over clouds affects clear-sky fraction

Comparison of the CAFS-ATom measurements of J-values with modeled ones presents a fundamental disconnect, but one that we must work through if we are to test the J-values in our chemistry-climate models with measurements. A CAFS observation represents a single point with unique solar zenith and azimuth angles within a unique 3D distribution of clouds and surface

albedos. Ozone column and temperature also control J-values but are less discontinuous across flight path and model grids. One can define a column atmosphere (CA) for each J-value in terms of the clouds directly above/below and the surface albedo, as would be measured by satellite nadir observations. The CAFS measured actinic flux includes direct and diffuse light, which depends on all the neighboring CAs out to 10s of km. Adjacent clouds can either increase or decrease the scattered sunlight
at the measurement site depending on location of the sun.

By including cloud fractional coverage from the meteorological models, and attempting in various ways to describe cloud overlap, the models here recognize that the atmosphere is not horizontally homogeneous. Yet, for cost effectiveness and non-random J-values, most modeling solves the radiative transfer problem for a 1D plane-parallel atmosphere that is horizontally
homogeneous. Most chemistry models adopt a simple averaging procedure to create a single, horizontally homogeneous cloudy atmosphere in each grid cell and then solving for a single J-value (see Table 1). The UCI model uses decorrelation lengths to determine cloud overlap, to generate a set of independent column atmospheres (ICAs), to generate 4 quadrature column atmospheres (QCAs), to calculate 4 J-values, which are then averaged to get a single J-value. In either case, the radiative transfer solution is 1D and there is one J-value per grid cell given to the chemistry module (and analyzed here).

What would the probability distribution rlnJ look like if we used the UCI J-values from the QCAs before averaging? For this, we collect the statistics on total COD for the two geographic blocks and compare the sub-grid QCA CODs against averaging approaches in Figure 6. This histogram (blue dots) is our best estimate of the distribution of total COD from a 1D nadir perspective of all the sub-grid ICAs. The UCI J-values are calculated as the average over 4 quadrature column atmospheres
(QCAs) as representatives of the ICAs. The two single-column atmosphere (SCA) models use simple averaging (COD x CF, green dots) and B-averaging (COD x $CF^{3/2}$, orange dots), which reduces the cloud fraction but assumes maximal cloud overlap. In these cases, a single J-value calculation is made with the SCA. When the clouds are simply averaged over the grid cell (~1° x 1°), the clear-sky occurrence drops to 7 % and the deep cumulus disappears. When Briegleb (1992) B-averaging is used, there is only slightly more clear sky (12 %). Both averaging methods also reduce the occurrence of thick cumulus. When run
at lower resolution, both averages find less clear sky, while the QCA statistics are not affected by resolution in the range 50-200 km. This averaging, of either J-values or clouds, explains why most models do not produce a single, sharp peak at rlnJ = 0.

At sufficiently high model resolution, where CF is either 0 or 1, a new problem arises because the radiative transfer problem
is now clearly 3D. The 1D radiative transfer used here would produce a very sharp clear-sky peak in rlnJ that is not seen in CAFS. The CAFS rlnJ distribution is widened in part by TUV albedo biases, but also because it is effectively a weighted average of cloud conditions over 10s of km or more and thus has lower frequency of clear sky than do the 1D ICAs. The Group 1 models with large peak distributions at rlnJ~0 appear to be basically incorrect since averages over these model

resolutions (>0.5°) should reduce clear-sky occurrence. There is probably a sweet spot in model resolution at about 20 km where the model statistics, even with 1D RT, should match the observed statistics.

## 4.4 Ocean surface albedo

We chose our Pacific blocks for this comparison to avoid large aerosol contributions and to be oceanic to avoid large variations
in surface albedo. Nevertheless, the ocean surface albedo (OSA) is variable (Jin et al., 2011), but most of these models, including TUV, assume a uniform low albedo in the range of 0.05 to 0.10. For the modeled ratio J-cloudy to J-clear, using a fixed albedo is not so important since both J-values use the same albedo. For the CAFS/TUV ratio, however, it is essential to have the TUV model use the OSA that best corresponds to the sea surface conditions under the CAFS measurement. Work on the CAFS/TUV calibration seeks to achieve this zero bias, and it continues beyond the cutoff date of the ATom-1 data used
here. The OSA affects our 2 J's differently: for J-O1D with peak photolysis about 305 nm, the OSA under typical conditions (SZA = 20°, surface wind = 10 m/s, chlorophyll = 0.05 mg/m3) is 0.038; while for J-NO2 with peak photolysis at 380 nm, the OSA is 0.048. OSA depends critically on the incident angle of radiation, increasing from 0.048 at 20° to 0.068 at 50° (380 nm). Rayleigh scattered light has on average larger incident angles than the solar beam for CAFS measurements and is reflected more than the direct beam. Rayleigh scattering is much more important for J-O1D than J-NO2.

The UCI standalone photolysis model was rewritten to include a lower boundary albedo that varies with angle of incident radiation, and is now designated Cloud-J version 8. In Cloud-J there are 5 incident angles on the lower surface: the direct solar beam and the 4 fixed-angle downward streams of scattered light. The ocean surface albedos (OSA) modules are adapted from the codes of Séférian et al. [2018] based on Jin et al. [2011], but we do not use their approximation for a single 'diffuse'
radiation since all of Cloud-J's scattered light is resolved by zenith angle. The resulting albedo is a function of wavelength, wind speed, and chlorophyll; it is computed for each SZA and the 4 fixed scattering angles. As a test of the importance of using a more realistic OSA, we used Cloud-J v8 to compute the ratio of J with our OSA module to J using a constant fixed albedo, with both J's calculated for clear-sky. We calculate this rlnJ (log of the ratio of J-clear (OSA) to J-clear (single albedo)) for a range of SZA, wind speeds, and chlorophyll comparing to fixed albedos of 0.00, 0.06 and 0.10 as shown in Figure S9.
Because of the range in conditions, there is no single offset, but we have a probability distribution whose width in rlnJ due to a range of surface albedo has a magnitude that affects the interpretation of measurement-model differences. For high sun (SZA = 0° – 40°), choosing an optimal fixed albedo of 0.06 results in little mean bias, although individual errors are about ±2%. This error is based on conditions for cos(SZA) > 0.8. If the fixed albedo differs from this optimum (e.g., 0.00 or 0.10), then bias errors of 2 % to 10 % appear, and the width of the distribution expands greatly. For SZA = 40° – 80°, the optimum
fixed albedo starts showing bias and has a much broader range of errors under different circumstances (wind, chlorophyll, SZA). Thus J-values calculated using unphysical, simplistic fixed ocean surface albedos can have errors of order ±10% depending on ocean surface conditions and the angular distribution of direct and scattered light at the surface. These errors

will not directly affect the model results here since the cloudy-clear differences used a self-consistent albedo in each model. Overall, however, there is a need for chemistry models to implement a more physically realistic OSA.

## 5 Discussion

The importance of clouds in altering photolysis rates (J's) and thence tropospheric chemistry is undisputed. On a case level it is readily observed, and on a global level, every model that has included cloud scattering finds significant changes in chemical rates and budgets. For example, Spivakovsky et al.'s (2000) inclusion of cloud layers caused a shift in peak OH abundance from the boundary layer to just above it, resulting in a shift to colder temperatures (and lower reaction rates) for the oxidation of $CH_4$-like gases, even with the same average OH abundance. Other than single-column, idealized, off-line tests of radiative transfer methods, we have few methods to constrain modeled J's under cloudy conditions.

The impact of clouds on J's is large, simply by looking at the profile of mean J's (Figures 1, 2, and 3); and it greatly complicates the comparison of J-values across models. The modeled mean clear-sky profiles, including CAFS TUV model (Figure 2), tend to agree within ±10% with the exception of MOCA, GISS and sometimes IFS. However, the all-sky (cloudy) profiles (Figure 1) have a much wider spread, except for J-O1D in the north Pacific for which there is inexplicably a core group of 8 models within ±10%. The observed CAFS cloudy profiles show a distinctly different profile in the north Pacific versus the tropical Pacific for both J-O1D and J-NO2 (Figure 1) and especially in the CAFS/TUV derived J-cloudy to J-clear ratios (Figure 3). Many models follow this typical profile in the tropical Pacific (100-900 hPa, Figure 3), but some (NCAR, UCI, UCIb) show large shifts to enhanced J-values immediately above the marine boundary layer. Whether this fundamental shift in cloud regimes is robust remains uncertain, and may be resolved with the full set of ATom deployments or with careful model studies, see below.

A more informative diagnostic of cloud effects is the statistical distribution of the individual J-cloudy to J-clear ratios (rlnJ). To model clouds correctly we need to understand how frequently and by how much clouds interfere with J-values. The statistical distribution of rlnJ shows distinct patterns and classes of models. In the free troposphere (100-900 hPa), all have a sharp edge at rlnJ = 0 with hardly any diminished J-values (rlnJ < 0), but the small-cloud-effects models (GC, GFDL, GISS, GMI, UKCA) show a dominant peak at rlnJ ~ 0 while the large-cloud-effects models (IFS, MOCA, NCAR, UCI, UCIb) have extensively enhanced J-values with large fractions at +25% or more. In the marine boundary layer (surface-900 hPa), this pattern is reversed with a sharp edge for all models at rlnJ ~ +0.03, and the large-cloud-effects models showing more diminished J-values. CAFS/TUV has its own pattern, which is very broadened, but generally supports the large-cloud-effect models. If further analysis can tighten the CAFS rlnJ distribution, then we may be able to discriminate among the models in one of the classes. One robust result from the models is that large cloud effects (|rlnJ| > 0.05) are always asymmetric, splitting in sign above and below 900 hPa. If the symmetric spread in CAFS rlnJ remains robust with additional deployments and

efforts to reduce CAFS-TUV inconsistencies, then we have a clear challenge for the cloud climatologies used in the chemistry models.

More work on the CAFS/TUV data could help better discriminate among the model classes identified here. For one, we need to sample over different seasons and synoptic conditions to build a more robust climatology. Fortunately, the additional three deployments (ATom-2, -3, -4) will provide these. They occur in different seasons; that is a consideration; but the tropical oceanic data can probably be combined. The analysis can be extended to the Atlantic. CAFS or similar measurements from other aircraft missions could also be added. Other tasks involve tightening the spread in rlnJ by looking for potential measurement-model "noise" (e.g., aircraft-sun orientation, sun-cloud geometries) and by improving the TUV clear-sky modeling (e.g., a more accurate ocean surface albedo derived from observed surface wind, chlorophyll, SZA). An unresolved issue is how to treat aerosols, both in the models and in CAFS. If 'clear sky' includes aerosols then TUV must be able to infer an aerosol profile for all measurements, and likewise the models need to be careful in how they calculate cloudy-to-clear. Fortunately for most of the oceanic ATom measurements, aerosol optical depths are small, but where they are large (e.g., Saharan dust events) the daily satellite mapping should provide adequate coverage for the TUV modeling. Developing these cloudy-to-clear J-value statistics over land will be more difficult due to the higher inherent variability in albedo and aerosol profiles.

More effort is needed from the modeling community to characterize the key factors driving these model differences in photolysis rates under realistic, cloudy conditions. This might include sensitivity runs that address aerosol and surface albedo impacts for each model. We would also need a better characterization of the cloud distributions used in chemistry models, including comparison with satellite climatologies (Cesana and Waliser, 2016; Ham et al., 2017), to understand how cloud fraction and overlap affects the J's used in the photochemical calculation of a column atmosphere. Models can help assess the ATom CAFS statistics for representativeness by checking on flight routing, multiple days, or different years.

The 3D nature of the radiation field measured by aircraft presents a more fundamental challenge. The observations average over cloud fields out to 10s of km, and even if the atmospheric column at the aircraft is clear, neighboring clouds alter J-values. The chemical models are coarse resolution compared with the CAFS measurements and average over wider range of cloud fields, almost eliminating the occurrence of clear-sky conditions. Even if they calculate a distribution of J-values for the cloud statistics in a column (like Cloud-J), they combine these to deliver a single average J-value for the chemistry module, again reducing the occurrence of clear-sky J-values. Thus, the result above (i.e., that 5 models have about 50% nearly-clear-sky J-values down to the surface) is inexplicable unless they have column optical depths of 0.3 or very small effective cloud fractions. At some model resolution, probably of order 10s km, the CAFS measurements may be a statistical representation over that grid; and our comparisons, even with 1D radiative transfer over a range of ICAs, may be more consistent. At these scales, there remains the problem of a strong zenith angle dependence (Tompkins and Giuseppe, 2007). Super-high resolution models

(~1 km) are becoming available in regional or nested-grid models (Kendon et al., 2012; Schwartz, 2014; Berthou et al., 2018), and one might hope that our problem is now solved because each single column atmosphere (SCA) explicitly resolves cloud overlap with each grid cell being either cloudy or clear. The calculation of photolysis and solar heating rates is not simplified, however, because now the SCAs interact with neighbors. To calculate the correct rates at any location, or even the average over a region, would require that we calculate the ratio of cloudy to clear over a 20-km domain of cloud-resolved grid cells.

**Data availability:** The data sets used here (global hourly J-values) for the plots and analysis are extensive (~10 GB) and will be archived at the Oak Ridge National Laboratory DAAC. This includes also the 3-second CAFS from all ATom-1 research flights. See Hall et al. (2018), https://doi.org/10.3334/ORNLDAAC/1651. The data analysis and plotting codes, as MATLAB scripts, are archived there also.

**Author contributions**: SRH and MJP designed this analysis. MJP developed the codes to analyse, compare and plot the data. SRH and KU performed the CAFS analysis and contributed those data. All other others ran the prescribed model experiments and contributed their model data. MJP and SRH prepared the manuscript with review and edits from all other co-authors.

**Competing interests:** The authors declare that they have no conflict of interest.

### Acknowledgments

This work is supported by the ATom investigation under National Aeronautics and Space Administration's (NASA) Earth Venture program (grants NNX15AJ23G, NNX15AG57A, NNX15AG58A, NNX15AG71A), Earth Science Project Office (see https://espo.nasa.gov/atom/content/ATom), and the Oak Ridge National Laboratory Distributed Active Archive Center (see https://daac.ornl.gov/ATOM). The National Center for Atmospheric Research is sponsored by the National Science Foundation. The UKCA simulations used the NEXCS High Performance Computing facility funded by the U.K. Natural Environment Research Council and delivered by the Met Office. ATA and NLA thank NERC through NCAS and through the ACSIS project which has enabled this work. VH acknowledges funding from the Copernicus Atmosphere Monitoring Service (CAMS). AMF acknowledges L.W. Horowitz (NOAA GFDL) for technical guidance with AM3.

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

**Table 1. Modeling photolysis and cloud fields**

| short name | long name | Cloud data (resolution) and date | J-values and cloud fraction (CF) treatment | Model references, including J-values |
|---|---|---|---|---|
| GC | GEOS-Chem | Cloud CF+OD from MERRA-2; GC v11_01. (2.5°x2.0°). 2013 Aug 16 | Fast-J* v7.0, single column, Briegleb averaging** | Gelaro et al., 2017; Liu et al., 2006, 2009. |
| GFDL | GFDL AM3 | Cloud data from 0.5° AM3 using 1.4° NCEP (u,v). (0.5°x0.5°) 2013 Aug 16 | Fast-J v6.4, liquid cloud C1 (12 μm) and ice clouds per Fast-J, Briegleb averaging | Donner et al., 2011; Naik et al., 2013; Mao et al., 2013; Li et al., 2018; Lin et al., 2012. |
| GISS | GISS Model 2E1 | Clouds from climate model nudged to MERRA fields (2.5°x2.0°) 2013 Aug 16 | Fast-J2. | Schmidt et al., 2014; Shindell et al., 2012; Rienecker et al., 2011. |
| GMI | GSFC GMI | Cloud CF+OD from MERRA-2 (1.3°x1.0°) 2016 Aug 16 | Fast-J v6.5, liquid cloud C1 (6 μm) and ice cloud hexagonal (50 μm), Briegleb averaging | Strahan et al., 2013; Duncan et al., 2007. |
| IFS | ECMWF IFS | Cloud data from IFS (0.7°x0.7°) 2016 Aug 15 | Williams et al. (2012). Liquid cloud (4-16 μm, using CCN), ice clouds, random overlap. | Flemming et al., 2015; Sun and Rikus, 1999; Sun, 2001. |
| MOCA | MOCAGE | Cloud data from ARPEGE operational analysis, 3 h. (1.0°x1.0°) 2017 Aug 16 | From Brasseur et al, (1998), using CF and liquid water (10 μm), Briegleb averaging. | Guth et al., 2016; Arteta & Flemming, 2015 |
| NCAR | CESM | Clouds from CAM5 physics on MERRA (u,v,T, …). (0.6°x0.5°) 2008 Aug 16 | TUV lookup J-tables, scaled using CF and liquid water content; Briegleb averaging. | Tilmes et al., 2016; Madronich, 1987 |
| UCI | UCI CTM | Cloud data from IFS T159L60N160 forecasts by U. Oslo. (1.1°x1.1°) 2005 Aug 16 | Cloud-J v7.3, quadrature column atmospheres from decorrelation length. Liquid and ice clouds per Fast-J. | Neu et al., 2007; Holmes et al, 2013; Prather 2015; Prather et al., 2017 |
| UKCA | UKCA | Cloud data from UK Unified Model (1.9°x1.3°) 2008 Aug 17 | Fast-J v6.4, cloud optical depths per Telford et al (2013). Briegleb averaging. | Morgenstern et al 2009; O'Connor et al 2014; Walters et al 2017. |
| UCIb | UCI CTM | same as UCI | Cloud-J v7.3, single column, Briegleb averaging. | |

Cloud data includes: cloud fraction (CF), in-cloud ice/liquid water path and effective radius, or in-cloud ice/liquid optical depth (OD in the visible).

*Fast-J versions here based on Bian and Prather (2002) with updates, including standard tables for cloud optical properties and simplified estimate of effective radius. Cloud C1 refers to Deirmendjian liquid cloud size distribution from the Fast-J data tables (Wild et al., 2000).

**Briegleb's (1992) method approximates maximum-random overlap with a single column atmosphere and adjusted effective cloud fraction such that the cloud optical depth in the grid cell is COD(in-cell) = COD(in-cloud) x $CF^{3/2}$.

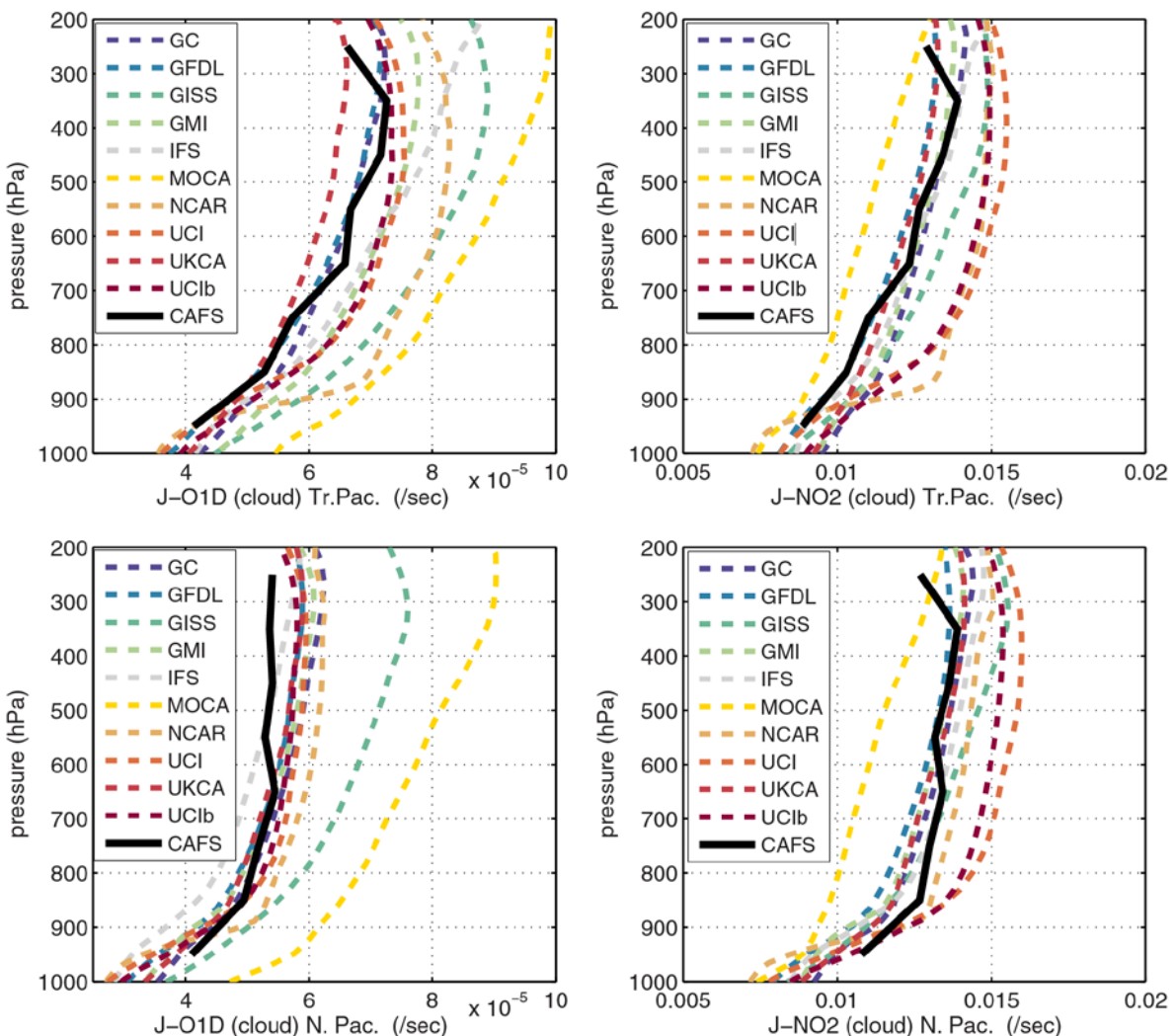

Figure 1. Profiles of all-sky ('cloudy') J-O1D and J-NO2 for the Tropical and North Pacific blocks. See Figures S2 and S3. The CAFS values are directly measured in ATom-1. The 10 models are sampled over 24 hours from a day in mid-August, selecting for cos(SZA) > 0.8. The UCI and UCIb models are distinct here because they treat overlapping clouds differently (cloud quadrature versus B-averaged cloud).

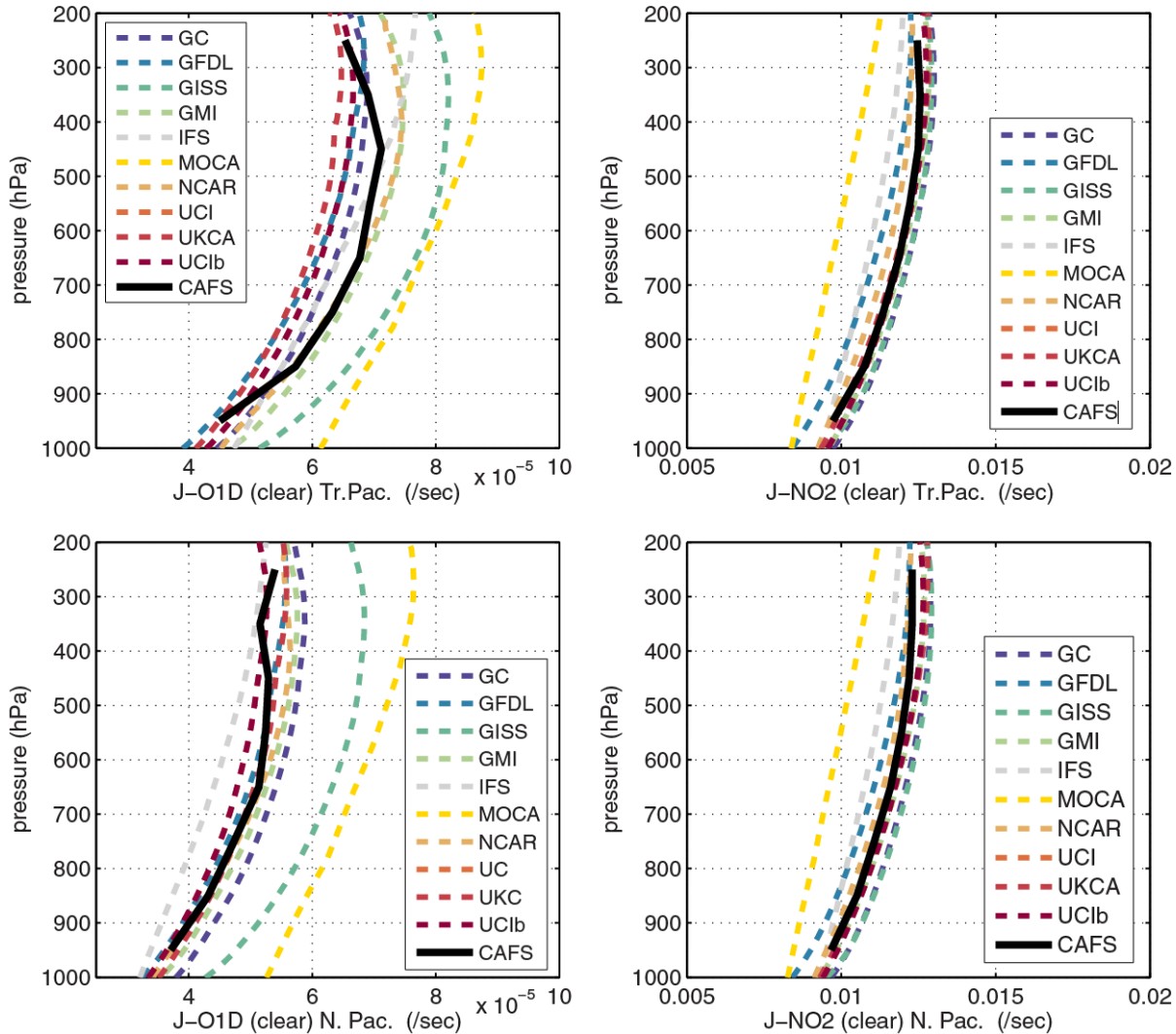

Figure 2. Profiles of clear-sky J-O1D and J-NO2 for the Tropical and North Pacific blocks. See Figure 1. CAFS here refers to TUV J's modeled at each point along the flight path. The UCI and UCI models are not separable since both have the same clear-sky J's. The spread in J-NO2 is likely due to different choices for interpolating cross sections and quantum yields. The J-O1D spread may be caused by the different ozone columns in the tropics, see Figure S3.

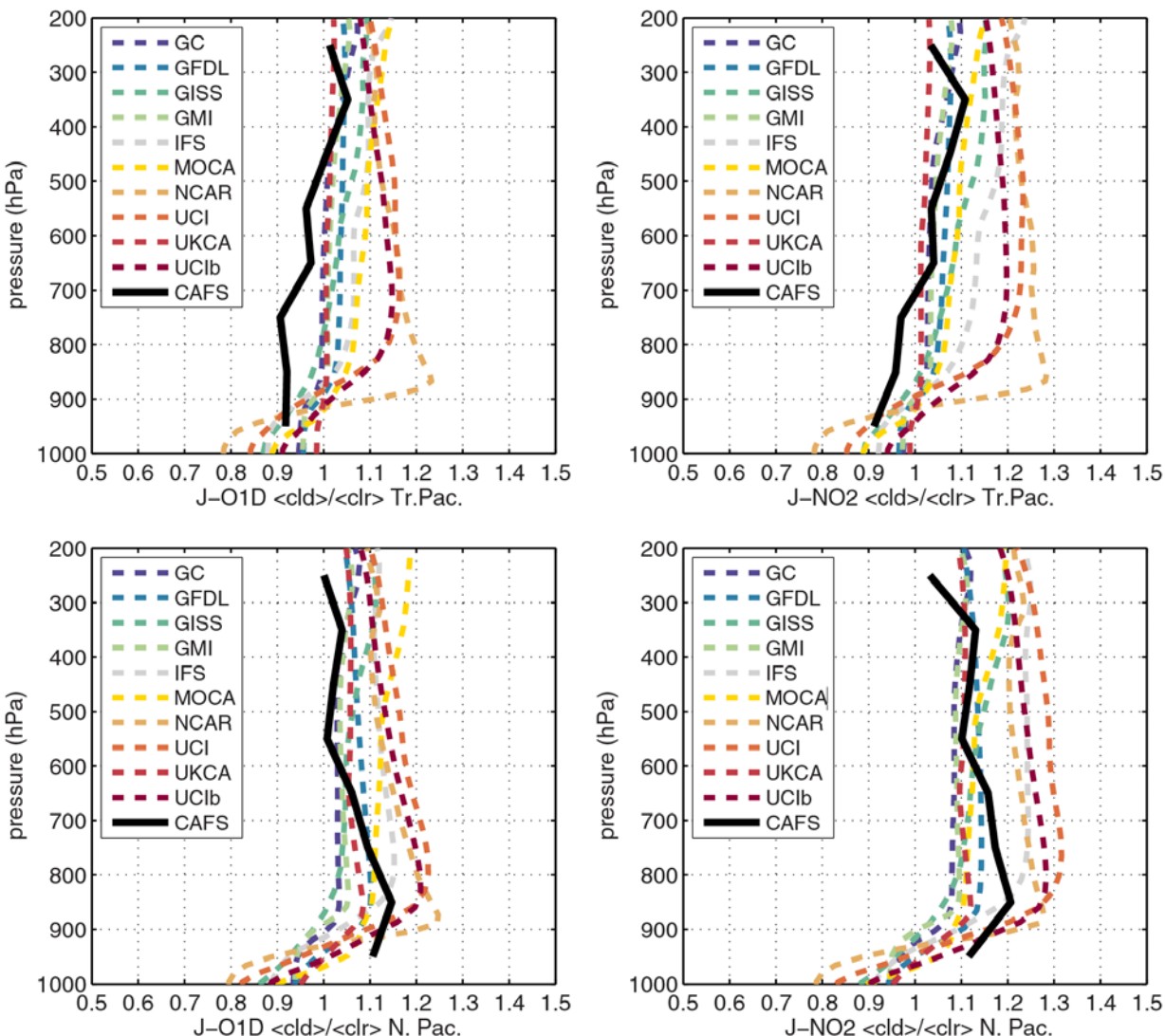

Figure 3. Profiles of the ratio of the average of J-cloudy to the average of J-clear for J-O1D and J-NO2 and for the 2 Pacific blocks. See Figures 1 and 2.

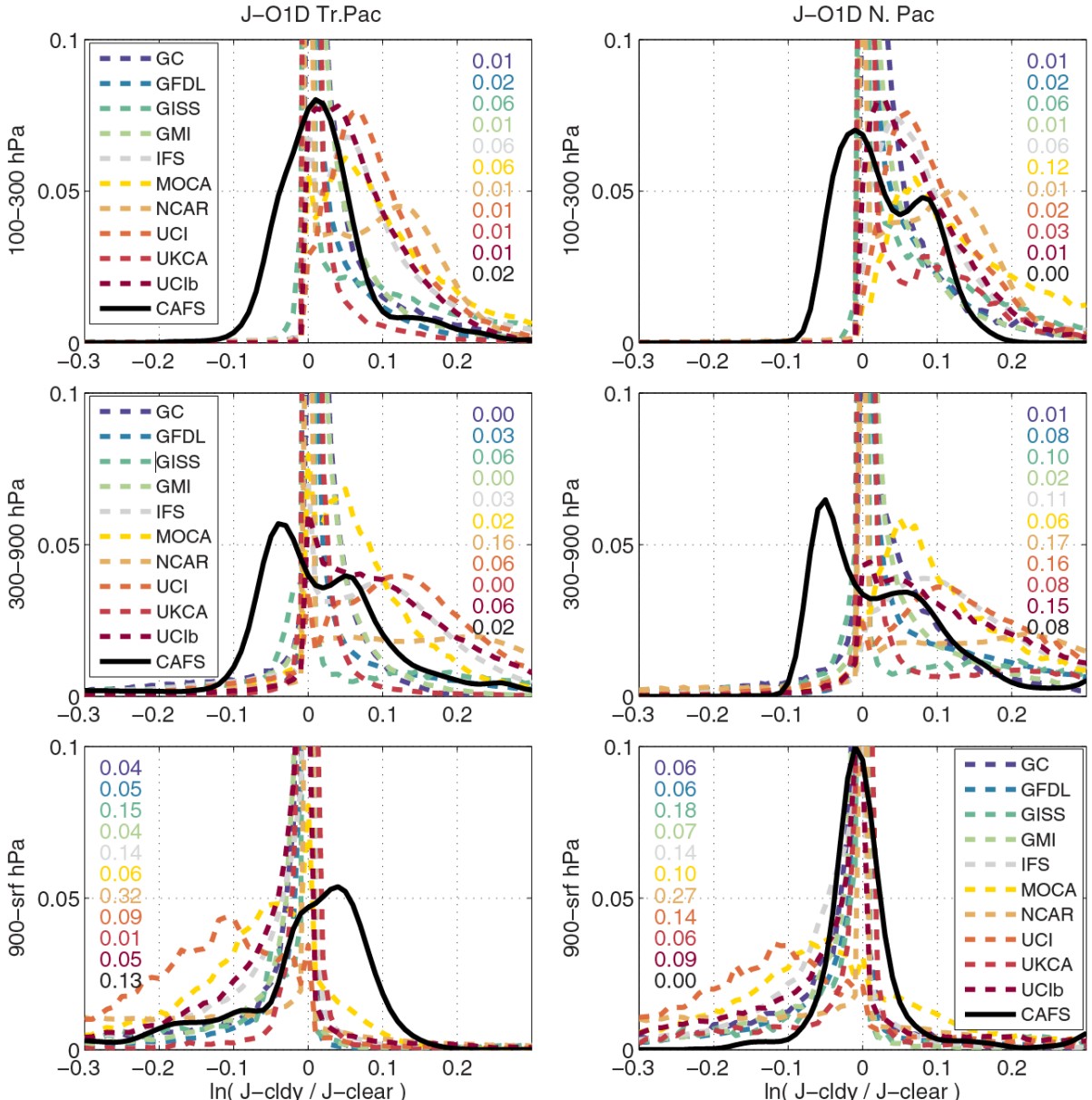

Figure 4a. Probability distribution of the natural log of the ratio of cloudy-to-clear J-O1D values (rlnJ) from 10 models and from CAFS during ATom-1. The columns correspond to the 2 geographic blocks (Tropical Pacific, 20 ºS – 20 ºN x 160 ºE – 240 ºE, and North Pacific, 20 ºN – 50 ºN x 170 ºE – 225 ºE). The rows are the 3 pressure layers (100 – 300, 300 – 900, 900 – surface hPa). All histograms sum to 1, but for many models the peak values about rlnJ = 0, corresponding to cloud-free skies, are truncated. Where a significant fraction of events does not fit within the ±0.3 range – on the high side for 100 – 900 hPa and low side for 900 – surface hPa – the column of numbers, placed on the appropriate side and color coded to the legend, gives the fraction of occurrences outside the range.

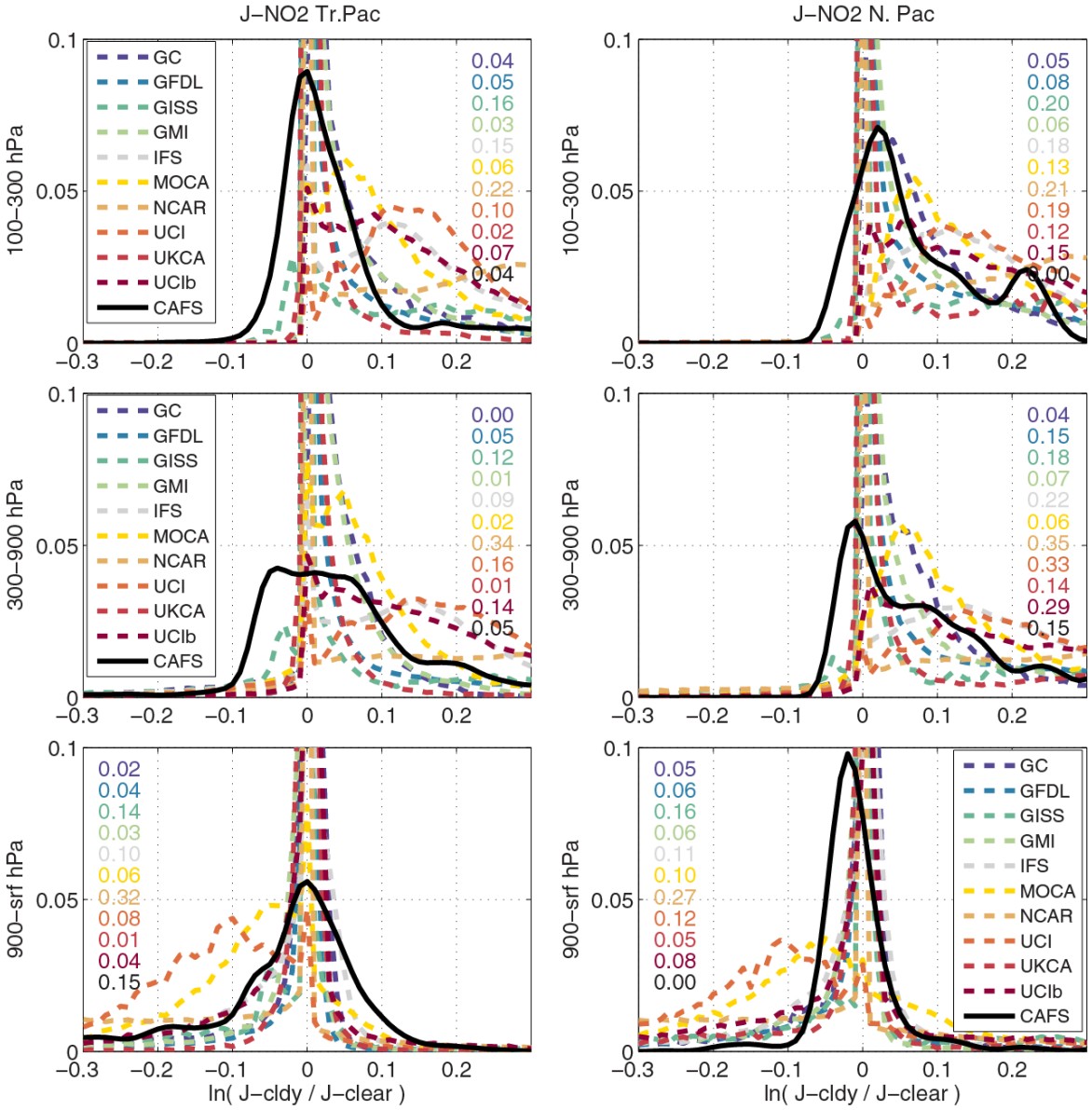

Figure 4b. Probability distribution of the natural log of the ratio of cloudy-to-clear J-NO2 values (rlnJ) from 10 models and from CAFS during ATom-1. See Figure 4a. In general, J-NO2 is more response to clouds than is J-O1D.

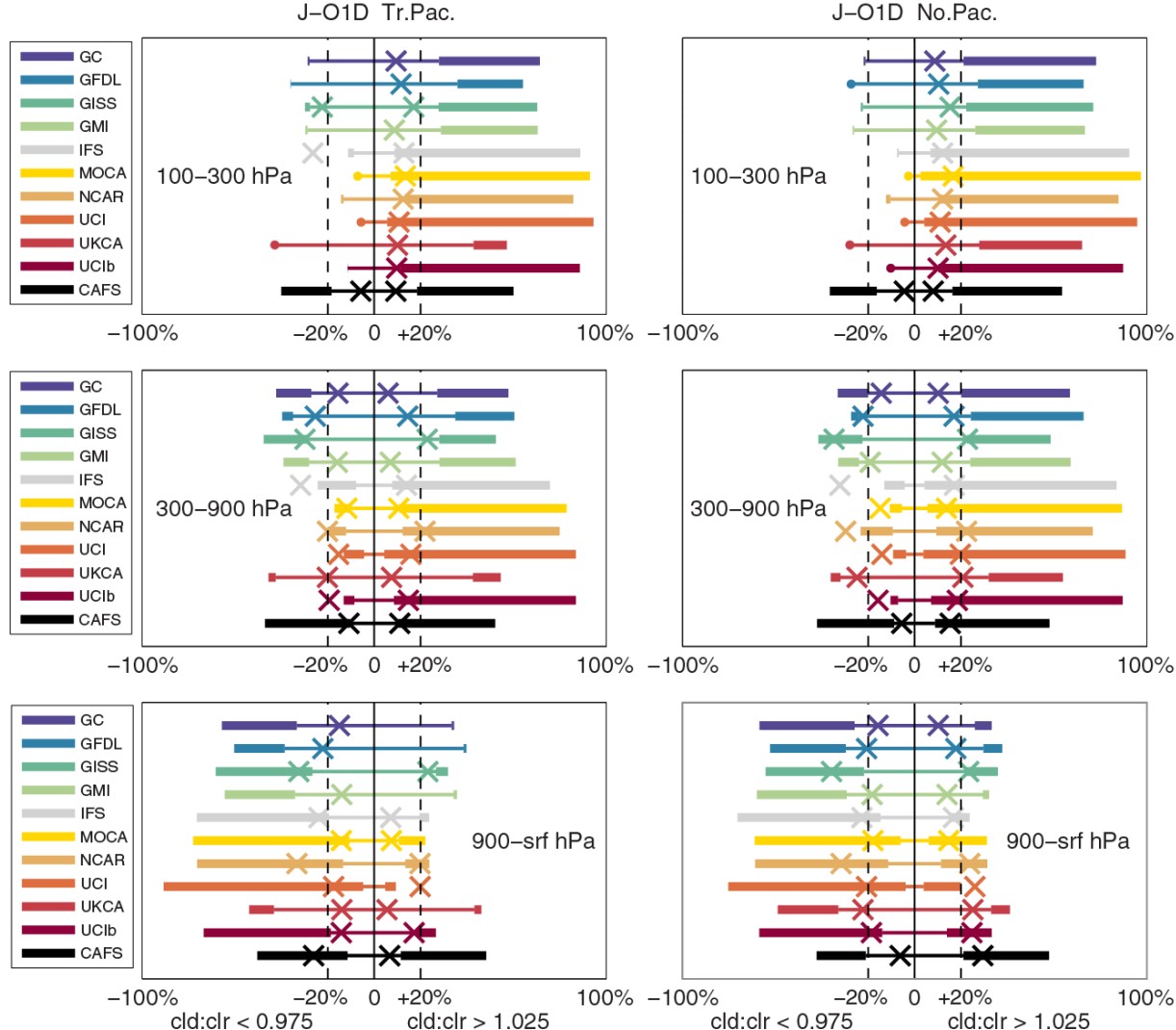

Figure 5a. Frequency of occurrence and magnitude of change in J-O1D caused by clouds. The panels and data sources are the same as in Figure 4ab. The horizontal lines all have length 1 and show the fraction of (i) cloud-diminished J's (thick left segment, cloudy:clear ratio < 0.975), (ii) nearly cloud-free J's (thin central segment, 0.975 < ratio < 1.025), and (iii) cloud-enhanced J's (thick right segment, ratio > 1.025). Each line is plotted with its nearly-cloud-free segment centered on 0. The mean magnitude of diminishment/enhancement corresponding to the thick line segments is plotted as an 'X' on each line segment, using the X-axis [-1, +1] as the natural log of the cloudy:clear ratio. Ratio changes of -20 % and +20 % are shown as dashed vertical grid lines. The 'X's are not shown when the frequency of occurrence of either thick segment is < 0.02. For an example of how to read these figures consider the panel in row 3 column 1 (J-O1D, Tr. Pac, 900 – srf). The GFDL model has about 22 % cloud-diminished J's (left segment) with an average value of 22 % below clear-sky J's (the 'X' on the left side); most of the remaining, 76 %, are nearly cloud-free (central thin segment). The GISS model has (from left to right) 42 % cloud-diminished J's, 53 % cloud-free J's and only 5% cloud-enhanced J's for a total of 100 %; the cloud-diminished J's average about 28 % less than clear-sky J's (the 'X' on the left) while the cloud-enhanced J's average about 22 % greater (the 'X' on the right side). See legend in Figure 5b.

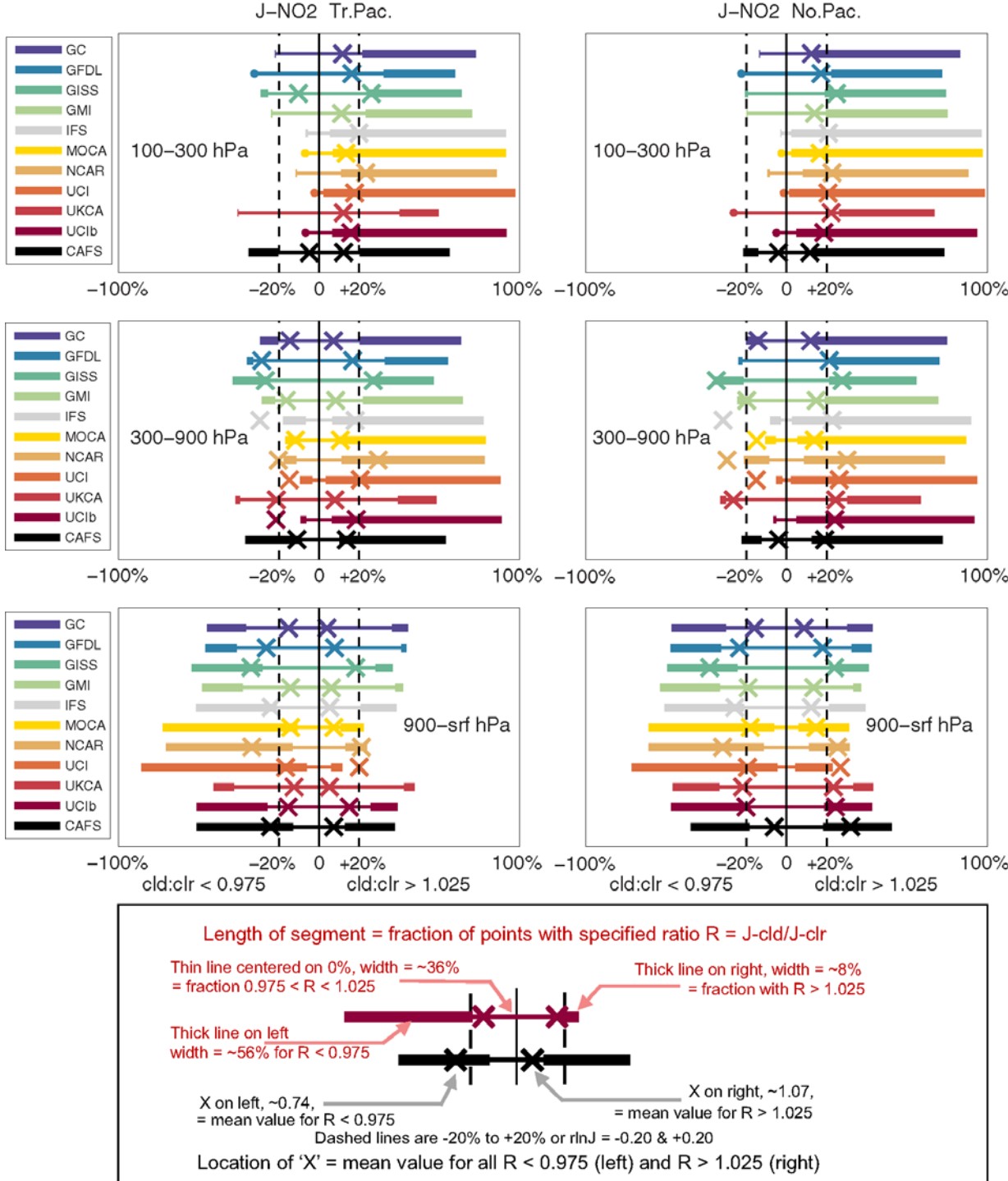

Figure 5b.  Frequency of occurrence and magnitude of change caused by clouds in J-NO2. See Figure 5a.

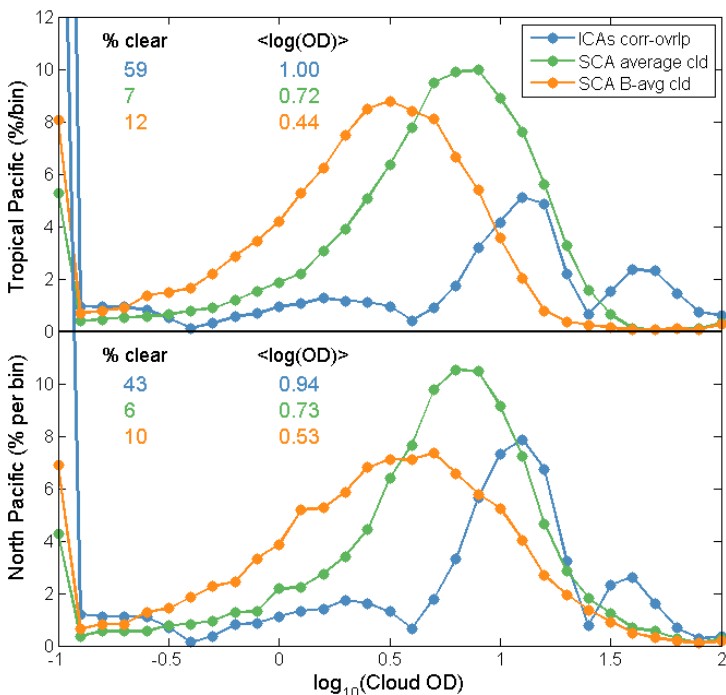

Figure 6. Histogram (% per 0.1 bin) in log of the total cloud optical depth (COD) based on method of implementing fractional clouds: Cloud-J independent column atmospheres (ICAs, blue dots); simple cloud averaging over each cell (COD x CF, green dots) and B-averaging (COD x $CF^{3/2}$, orange dots). The 600-nm COD for two large regions (Tropical Pacific, 20 ºS – 20 ºN x 160 ºE – 240 ºE, and North Pacific, 20 ºN – 50 ºN x 170 ºE – 225 ºE. is collected for 16 Aug 2016 from eight 3-hour averages of COD and cloud fraction (CF) in each model layer. The "% clear" is the sum of fractions (%) with $\log_{10}$(total COD) < -0.5; and the "$<\log(OD)>$" is the average of $\log_{10}$(total COD) > -0.5. The cloud fields come from the ECMWF IFS cycle 38 system run at T159L60N160 resolution. On average the number of ICAs per cell in the tropical block is about 170, although individual cells may have >1000. In spite of the high, 40-60 % fraction of "clear" columns, the quadrature-averaged J-values usually include some cloudy fraction.