# Peer review of "Cloud impacts on photochemistry: building a climatology of photolysis rates from the Atmospheric Tomography mission"

_Atmospheric Chemistry and Physics, 2018_

## Referee Comment (RC1) · Anonymous Referee #3 · 14 Oct 2018

This paper describes the novel use of observation-based photolysis rates derived from aircraft profile flux measurements to test global atmospheric chemistry models. It adopts an original and innovative approach, and provides a highly valuable first step in the evaluation of cloud impacts on photolysis rates in models, deriving some useful pointers to areas that need further exploration or model development. The study is well executed and thorough, exploring sensitivity to ozone column and albedo as well as to a central focus on cloud properties. The analysis approaches appear sound, the conclusions are useful, and as such the paper merits publication in ACP once some minor issues have been addressed.

[Figure]

Specific comments

Data was provided from the models for a single day in August. How dependent are the model results on the specific day chosen? Are the regions large enough to provide a truly representative distribution of cloud coverage? This would be relatively simple to check. In addition, the number of observed samples per block is stated on page 5, but not the number of low-SZA model samples.

The coarse resolution of the models leads to an averaging of cloud cover and an underestimation of clear sky conditions (explored in section 4.3). What would the distribution of rlnJ look like for the observations if these were averaged to the physical scale of the models?

To what extent does the noise inherent in the observation-based rlnJ values (evident in the broadening seen in Fig 4) wash out the vertical profile of cloud impacts on photolyis rates that is clearly seen in the models? Does the occasional enhancement of near-surface J-values and reduction at high altitudes reflect biases in model cloud distributions, 3-D effects, or just noise?

Is Fig 2 based on additional model runs without clouds (alluded to on page 4, line 23) or on clear-sky columns that are a subset of the all-sky data shown in Fig 1? This should be stated on page 6. If the clear-sky values are a subset, what bias does the different locations and SZAs of clear and cloudy columns introduce?

Fig 5 is an interesting and well thought out way of presenting the data, but is difficult for the reader to interpret, particularly where the bar indicating the proportion of reduced/enhanced J-values does not align with the cross indicating the mean magnitude.

Minor issues and Typos

Abstract, line 30: "more importantly" Please rephrase this or reorder the sentence appropriately.

Page 9, line 25: 3/2 should be a power

Page 10, line 23: section 4.3 is important, but the title of the section is awkward, please consider rephrasing "finding clear sky".

Page 11, line 6-7: this partly repeats information provided on page 5.

The caption to Figure 6 is too long. The information is valuable, but some of it should be included in the text on page 11. (Typo: please replace star with multiplication sign for consistency)

Caption to Fig S2, one line from bottom: "in the in the"

Caption to Fig S6, line 2: remove second occurence of "blocks"

Fig S6 needs to be cleaned up so that the titling is legible and the model legend is not superimposed on the axis labels.

---

## Referee Comment (RC2) · W.J. Collins (Referee) · 17 Oct 2018

Review of Hall et al. 2018

This paper nicely describes J-value measurements as part of the Atom-1 campaign. These measurements and those from further Atom campaigns will provide an excellent resource for assessing chemistry models.

The overall conclusion seems to be that "A primary uncertainty remains in the role of clouds in chemistry...". There should be enough information available to the authors to make some more conclusive statements. The authors need to think more about what

are the key messages they want people to take away from reading this paper.

The paper does not appear very clear as to its main purpose. Model-measurement comparisons can be used to assess whether models are fit for purpose (which doesn't seem the aim here) or to investigate where the models can be improved. This paper touches on the latter, suggesting that there are deficiencies in the treatment of clouds, but not what. It could be that models have too much or too little cloud, or that the frequency pdf of cloud amount is wrong, or that the radiation codes don't treat overlaps correctly or the parameterisation of cloud scattering is wrong. From this paper we still have no idea where to start looking to improve the models. Given the very different results between the observations and (some of) the models it must be possible to give some more concrete statements.

A very serious deficiency of the paper is the lack of coincident cloud observations. Cloudy/clear ratios are presented but with no data on what amounts of cloud caused these, it is therefore impossible to know whether the model-measurement differences are caused purely by different clouds. Measurements over 4 days in one particular August cannot be considered a representative climatology. Were these 4 days more or less cloudy than the climatological average? Where would the observed clouds for these 4 days lie on figure S6? Even if clouds weren't specifically measured by the Atom-1, cloud data will be available from satellite measurements or re-analysis data. This will require a significant amount of work by the authors, but I don't see the value of the paper without this.

A significant improvement to the paper would come from plotting the cloudy/clear ratios as a function of cloud fraction (at different levels) for both the models and observations (or reanalysis) – or the authors may come up with a better way of controlling for cloud amount. This would overcome a lot of the issue of likely very different clouds amounts in the 4 particular observed August days with 1 random modelled August day. It would identify whether for the same cloud amount, models and measurements were calculating very different J-values.

Abstract: This needs to contain the key messages from the study.

Page 1: This page seems mostly trying to justify why it is not sensible to do point comparisons. While the real world is 3D, the model radiation codes are typically 1D and so comparing the 1D models with 3D observations would actually seem a sensible test of whether 1D models can represent the real world. More importantly, the claim that 4 days of measurements can be considered a climatology representative even of August 2016 let alone Augusts in general is never questioned. No evidence is presented as to how representative the observed statistics of clear/cloudy might be.

Page 2, line 8: "... net probabilistic distribution of observed J-values". Again this glosses over the issue that the pdfs might be specific to those particular 4 days in August 2016, rather than being a more general climatological probabilistic distribution.

Page 5, line 16: Just as 4 days of observations can't be considered a climatology, a single day in August can't be representative. It should be possible to get much more than a single day's data from these models. Most of the models seem either to use re-analysis or were nudged; did they use meteorological fields from the days of the campaign or a random August?

Page 7, line 19: Are the narrow peaks due to the frequency of clouds? It could be that some models have an average cloud fraction of 0.1 by having a fraction of 0.1 every-where, but others have the same mean fraction with a large variety of cloud amounts. PDFs of CF and/or COD are needed (along the lines of fig 6).

Section 3.2: Given the large differences between the models it is essential to use more than the averaged cloud properties to understand the correlations between CF, COD and J-values, and how these differ between the models. How is COD calculated – does it use the same radiation scheme as for J-values or is from the climate radiation scheme?

Section 4.3: It would be useful to use UCI to replicate figure 4 for the ICAs and QCAs

**[ACPD](javascript:void(0))**
* * *
Interactive
comment

to determine how much the averaging contributes to the different distributions in observations and models.

Conclusions: This paper needs a conclusion section. As it stands it is not at all obvious what the overall conclusion of the study is, other than clouds are difficult and more work is needed.

---

## Referee Comment (RC3) · Anonymous Referee #2 · 19 Oct 2018

This paper presented a statistical analysis of the photolysis rates data obtained over the Pacific during the first deployment of the ATom mission (July 29-Aug.6, 2016) and evaluated the performance of nine global models with respect to how clouds affect photolysis rates (J-values). The ATom J-values are a unique data set for testing how well clouds are treated in terms of their impact on photochemistry in current global models. The model J-values presented in this study are from simulations with each model's own configuration without controlling input conditions among all nine models. Comparisons of the probability distributions of cloud impacts on J-values between the models and measurements provided original insights into model performance differences. The paper concludes with an interesting discussion on the difficulties and challenges involved

in simulating cloud impacts on J-values and comparing with measurements. The need for more effort to characterize the main factors contributing to the model differences are also acknowledged and discussed. The paper is overall well written and I recommend publication on ACP after some minor modifications.

Title: The title would suggest this paper is about the analysis of photolysis rates from all ATom campaigns, but in fact only the data from ATom-1 (July 29-Aug.6, 2016) is used to test nine global models. The title should reflect these. Also "climatology" (a term used throughout the text) cannot be derived from a single deployment. "Statistics" is probably a more appropriate term as part of the title for the kind of analysis presented, and is often used in the text.

Abstract: The abstract could be improved: 1). it doesn't flow that well. 2). P1, L28: "during the first deployment (ATom-1) in August 2016": it's actually July 29-August 6, 2016. 3). P1, L29: it would be useful to the reader to state that models provided hourly J-values for a single day of August for the domains measured in ATom-1. 4). L30: what is the statistical picture of the impact of clouds on J-values established by the ATom-1 measurements? 5). L31-32: "the models show largely disparate patterns", "there is some limited, broad agreement": what are the disparate patterns and broad agreement, specifically?

P4, L18: in Table 1

P5, L14-16: "It was not possible to have all the models simulate the flight paths and times" - why? Do most of these models use either assimilated or nudged meteorology? "we are trying to develop a climatology" - do you mean "statistics"? "the models were asked to pick a single day in August as representative of the cloud statistics over the large geographic blocks" - A single day of August 1-31, 2016, or near the end of the deployment (prior to Aug.6)? Figure S6 caption mentioned "one day in mid-August". Indicating the selected dates in the "Cloud data" column of Table 1 would help the readers who may want to reproduce or compare the results.

P6, 2nd paragraph: It's worth mentioning and explaining why the GC and GMI models, both driven by the MERRA-2 reanalysis, show large differences in JO1D over the tropical Pacific (upper left panels, Figs. 1-2).

P9, L11: the sharp peaks

P9, L25: superscript "3/2"

P13, L9, L14: "an observed climatology", "this CAFS climatology" - see above about "climatology".

P20, Table 1: GC "Cloud data" - the model uses liquid and ice cloud optical depths (not liquid and ice water) taken from MERRA-2. GC "Model references" - Gelaro et al. (2017) is the reference for MERRA-2. Cite Liu et al. (JGR 2006, 2009) for GC J-values and B-averaging.

P20, Table 1: 4th column, GFDL and GMI - what does C1 mean here? GMI "Cloud data" - the model uses liquid and ice cloud optical depths (not liquid and ice water) taken from MERRA-2.

Fig. S1 caption: low SZAs, not high SZAs

Fig. S2 caption: "in the in the"

Fig. S6 caption: Averaged in-cell cloud optical depth (COD, at 500-600 nm and .....); "blocks studies here blocks".

---

## Author Comment (AC1) · 1 Nov 2018

**Discussion of Big-picture Topics**, followed by the individual reviews and our responses.

*"climatology"*
This appears to be a loaded word, and we have used it too carelessly. We have shifted a lot of "climatology(ies)" to "statistics" as recommended, but some use of the word is retained since it is the intent to build such a climatology. Thus the title has been changed to "building a climatology." The models are tested as to the robustness of their statistics in terms of space, but we did not go back and ask for full resubmissions with different years. We have previously shown that sampling along a single meridian in the tropical Pacific produces a robust 2D distribution of chemical species (see Fig 8 below from the Prather et al: Atmos. Chem. Phys., 17, 9081–9102, 2017, https://doi.org/10.5194/acp-17-9081-2017), and have added a simple test here by splitting the tropical Pacific block into east and west halves.

*ATom-1 deployment only:*
One review was critical on the choice of dates and wished to put the "first deployment" with dates in the title. We tried several versions and it was just too clumsy, and not really title material. Thus we have brought forward in the abstract and paper the fact that we are only presenting ATom-1 data. The dates are now specific: the ATom-1 deployment consisting of 10 research flights was 29 Jul to 23 Aug 2016. While the full ATom-1 measurements of J-cloudy and J-clear are reported and archived with this paper, we analyze here only the Pacific data north of 20ºS, which includes research flights 1-5 (29-Jul to 8-Aug). Likewise, the current model analysis includes only the two Pacific blocks from 20ºS to 50ºN, but the archived model data is global and for a mid-August day (varying years). Thus, later analyses can look at the Southern or Atlantic Oceans (or continental data) using the archived data sets.

*Sampling of different years, more extensive cloud statistics, repeatability of just one day in August:*
We agree with reviewer Collins that it would be wonderful to have done this analysis for the full month of August and for different years to see if the modeled J-values statistics were robust and did not shift. Given the scope of the current work, and the effort from the modeling community, that would not have been possible, and we would have lost participants. As it was, and including some mistakes in diagnostics, the models often had to be corrected and re-run, and the amount of data processed was >60 GB.

   The reason for selecting large geographic blocks and 24 hours of model data were to make more robust statistics. Given the limits on available data, we have gone back to the J-values here and sub-sampled our declared "tropical Pacific" region into a west and east half, and also into a narrow stripe down the dateline. This was a useful exercise and the new results are presented for all models in Supplementary figures and discussed in the main text. Basically, most models show no difference across the sub-sampling (NCAR, GFDL, IFS, UCI, UKCA, MOCA) while three models (GC, GMI, GISS) show weaker cloud effects in the East half of the region. This is consistent in that these three use MERRA cloud fields in one way or another.

   There was some wish for further analysis of the cloud fields (which would require ice and liquid water paths, effective radii and cloud fraction) and their correlation with J-value anomalies. This new study would be interesting but require much more extensive work from the modelers, and of course, more detailed comparisons of how the models implement the cloud data. This would be better as a model-model set of fixed case studies.

*Why not just analyze the cloud fields?  or use assimilated/nudged cloud fields?*
We chose not to focus on comparing the modeled and observed cloud fields because they really do not tell us how J-values are perturbed by clouds.  Moreover, for the observations, there are no observed cloud fields at the resolution of the airplane measurements. J-values vary at the 3-sec resolution of the data (0.8 km) and 3D cloud fields at this scale are not available for the flight from either the aircraft or satellite.  There are some marvelous 3D high-resolution cloud fields (on the scale of the aircraft variability), but these are limited to CloudSat-CALIPSO overpasses that have been extended to 200-km wide swaths (see Barker et al., 2011; Miller et al., 2014), not to the 2,000+ km swaths needed to get daily coverage.   Even once-a-day coverage will not define clouds over the ATom flight.  Assimilating clouds is still a difficult, unsolved problem (from Alan Geer, ECMWF, 2017):

- forecast models still use simplified representations of cloud and convection
- the radiative transfer models required to simulate such 3D structures accurately would be far too expensive for operational use, and
- the predictability of cloud and precipitation tails off in a matter of hours.

A 2018 effort to assimilate clouds in high-resolution models (A.T. White. "Improving Cloud Simulation for Air Quality Studies through Assimilation of Geostationary Satellite Observations in Retrospective Meteorological Modeling", MWR, https://doi.org/10.1175/MWR-D-17-0139.1) is still based on the model physics that creates clouds: "The basic approach is to create positive vertical motion within the model to produce clouds and negative vertical motion to dissipate clouds, based on GOES cloud fields." So we cannot simply use the assimilated or 'nudged' cloud fields.  (Nudging uses U, V, T, q not clouds).

An additional reason why we do not spend more time on analyzing the cloud data from the models is that our primary objective is to find out how photochemistry, through J-values, is affected by clouds.  The way that models implement clouds can be very different, even when given the same key data: water path and effective radius for ice and liquid, and cloud fraction.  In the example pointed out by the reviewers, GC and GMI are both using MERRA clouds and should have similar results:  yet, they differ in clear-sky J's; but this difference become greater when we look at all-sky (cloudy) J's.  Our analysis here is not a thorough single-test-case model comparison in which one might be able to identify why the models take similar cloud data and use almost the same Fast-J code to give different cloud effects.  That would be interesting, but is beyond the scope of this paper.

Barker, H. W., M. P. Jerg, T. Wehr, S. Kato, D. P. Donovan, and R. J. Hogan, 2011: A 3D cloud-construction algorithm for the EarthCARE satellite mission. Quart. J. Roy. Meteor. Soc., 137, 1042–1058, doi:10.1002/qj.824.
Miller, S.D., J.M. Forsythe, P.T. Partain, J.M. Haynes, R.L. Bankert, M. Sengupta, C. Mitrescu, J.D. Hawkins, and T.H. Vonder Haar (2014) Estimating three-dimensional cloud structure via statistically blended satellite observations. J. Appl. Meteor. Climatol., 53, 437–455, doi:10.1175/JAMC-D-13-070.1

**Summary.**  Given that this paper is a start at analyzing how clouds control the statistical variation of J-values, it cannot answer all the questions we have.   The reviewers have come up with an interesting set of follow-on questions, which we would all like to see answered.  We believe that that we have presented a thorough, innovative approach for assessing how models implement clouds in their photochemistry, and come up with a new type of observational dataset to test this.  Some large questions remain as noted in the summary discussion (Section 5).  We believe the current content is extensive but necessary to understand this work.  Let this be the first step.

[Figure]

**Figure 8.** AIR-weighted 2-D probability distributions for $NO_x$ vs. HOOH averaged over tropical Pacific block (150–210° E, 20° S–20° N, 0–12 km) and for different single-longitude transects from 150–210° E, shown for models (**a**) C (GFDL-AM3) and (**b**) F (UCI-CTM). The fitted 2-D ellipses are shown for the full block (thick black line) and five longitude transects (colored lines) for models (**c**) C and (**d**) F. The block ellipse for the other model is shown as a thin black dashed line.

Anonymous Referee #3

This paper describes the novel use of observation-based photolysis rates derived from aircraft profile flux measurements to test global atmospheric chemistry models. It adopts an original and innovative approach, and provides a highly valuable first step in the evaluation of cloud impacts on photolysis rates in models, deriving some useful pointers to areas that need further exploration or model development. The study is well executed and thorough, exploring sensitivity to ozone column and albedo as well as to a central focus on cloud properties. The analysis approaches appear sound, the conclusions are useful, and as such the paper merits publication in ACP once some minor issues have been addressed.

*Thank you.*

Specific comments
Data was provided from the models for a single day in August. How dependent are the model results on the specific day chosen? Are the regions large enough to provide a truly representative distribution of cloud coverage? This would be relatively simple to check.

*This is a tough question. Please see big-topic answers at the beginning of this response. In part answer your question, we redid the sampling in the tropical Pacific to test the statistics (new Figure S7).*

In addition, the number of observed samples per block is stated on page 5, but not the number of low-SZA model samples.

*Yes, done. Some numbers were added in text, others in the figure caption.*

The coarse resolution of the models leads to an averaging of cloud cover and an under-estimation of clear sky conditions (explored in section 4.3). What would the distribution of rlnJ look like for the observations if these were averaged to the physical scale of the models?

*Interesting question. Using data and analyses similar to that for Figure 6, we looked at what happened as we took cloud data from a 640x320 (1/2 degree) model and averaged it over320x160 and 160x80. There was a detectable shift to less-clear skies, but the much bigger effect was on how the cloud fractions are used (Fig. 6). Thus, we deemed it not critical for this paper.*

*The second part of your question appears to ask about averaging the ATom-1 J's to a 100-km grid. There is no natural registration for such a grid and we did not try it. This would be better when we have added the last 3 deployments and have a larger data set. This topic is very important for model development: the best we can hope for is that modeled J's on 10-100 km scales are effectively the average of the 3D high-resolution fields (i.e., the ATom J's). This is discussed in Section 5.*

To what extent does the noise inherent in the observation-based rlnJ values (evident in the broadening seen in Fig 4) wash out the vertical profile of cloud impacts on photolyis rates that is clearly seen in the models?

*A very good, but difficult question to answer at this stage. This paper is based on the first released J-cloudy/J-clear ATom-1 data, and it points to the need for further work on*

*tightening up the spread (noise) in the distribution that could be due to correctable physical processes such as ocean surface albedo, airplane maneuvers (banking and ascent/descent can change the orientation of the 2-pi detectors relative to the direct beam of the sun), etc. When all 4 deployments are available, we can revisit this along with developing an improved set of model diagnostics.*

Does the occasional enhancement of near-surface J-values and reduction at high altitudes reflect biases in model cloud distributions, 3-D effects, or just noise?

*The low level enhancements are likely due to thinnish clouds where the J's are enhanced within and below. The low-level enhancement in observed J-O1D Tr.Pac. is just not sensible in terms of the 1D modeling, and this is now noted. We need to develop statistics on 3D cloud effects at the scale seen by the aircraft to truly answer this question.*

Is Fig 2 based on additional model runs without clouds (alluded to on page 4, line 23) or on clear-sky columns that are a subset of the all-sky data shown in Fig 1? This should be stated on page 6. If the clear-sky values are a subset, what bias does the different locations and SZAs of clear and cloudy columns introduce?

*Sorry for the confusion, this is rewritten on page 4 to be explicit: The models submitted two full 24-hour simulations: one with their standard cloud treatment and one where the aerosols and clouds were zeroed. No subsetting.*

Fig 5 is an interesting and well thought out way of presenting the data, but is difficult for the reader to interpret, particularly where the bar indicating the proportion of reduced/enhanced J-values does not align with the cross indicating the mean magnitude.

*Yes, Fig 5 is a dense figure in terms of information, yet all that information needs to be put together to understand the J statistics. We have added an explanatory legend to Figure 5b (it has the smaller figure caption) and explained more in the text about the 5 values being plotted for each model and the CAFS data.*

Minor issues and Typos
Abstract, line 30: "more importantly" Please rephrase this or reorder the sentence appropriately.

*Yes, done. Abstract is substantially rewritten in response to reviews.*

Page 9, line 25: 3/2 should be a power

*Yes, done.*

Page 10, line 23: section 4.3 is important, but the title of the section is awkward, please consider rephrasing "finding clear sky".

*Yes, done. Thanks."4.3 Averaging over clouds affects clear-sky fraction"*

Page 11, line 6-7: this partly repeats information provided on page 5.

*Yes, had dropped references and shortened a bit on page 11.*

The caption to Figure 6 is too long. The information is valuable, but some of it should be included in the text on page 11. (Typo: please replace star with multiplication sign for consistency)

*Agreed. Cut caption, added some discussion to text.*

Caption to Fig S2, one line from bottom: "in the in the"
*Yes, done.*

Caption to Fig S6, line 2: remove second occurrence of "blocks"
Fig S6 needs to be cleaned up so that the titling is legible and the model legend is not superimposed on the axis labels.
*Yes, both done.*

Anonymous Referee #2

This paper presented a statistical analysis of the photolysis rates data obtained over the Pacific during the first deployment of the ATom mission (July 29-Aug.6, 2016) and evaluated the performance of nine global models with respect to how clouds affect photolysis rates (J-values). The ATom J-values are a unique data set for testing how well clouds are treated in terms of their impact on photochemistry in current global models. The model J-values presented in this study are from simulations with each model's own configuration without controlling input conditions among all nine models. Comparisons of the probability distributions of cloud impacts on J-values between the models and measurements provided original insights into model performance differences. The paper concludes with an interesting discussion on the difficulties and challenges involved in simulating cloud impacts on J-values and comparing with measurements. The need for more effort to characterize the main factors contributing to the model differences are also acknowledged and discussed. The paper is overall well written and I recommend publication on ACP after some minor modifications.

*Thank you.*

Title: The title would suggest this paper is about the analysis of photolysis rates from all ATom campaigns, but in fact only the data from ATom-1 (July 29-Aug.6, 2016) is used to test nine global models. The title should reflect these.
Also "climatology" (a term used throughout the text) cannot be derived from a single deployment. "Statistics" is probably a more appropriate term as part of the title for the kind of analysis presented, and is often used in the text.

*See overview discussion on these 'big topics'. We now use 'statistics' more and 'climatology' less, but we still have the objective of developing a climatology for all models. Thus the title was changed to indicate the intent to build a climatology. It was impossible to put in all the information about the deployment in the title, and so we put it in the abstract and the opening sections.*

*Also the published data set (full ATom-1) and the analysis here are clearly stated in the abstract and intro.: "This analysis is limited to the first half of the deployment over the Pacific Ocean, but publishes the complete data set."*

Abstract: The abstract could be improved: 1). it doesn't flow that well.
*A substantial rewrite was done. Yes, it needed one.*

2). P1, L28: "during the first deployment (ATom-1) in August 2016": it's actually July 29-August 6, 2016.

*Yes, corrected. The ATom-1 deployment (in air) was 29 Jul to 23 Aug. The data from the tropical Pacific was taken between 29 Jul and 8 Aug (sorry for the 6 Aug typo in the manuscript). This is now accurately described in the paper. The model data is global and the ATom-1 CAFS data archived with this paper are complete, all 10 research flights, 29 Jul – 23 Aug 2016.*

3). P1, L29: it would be useful to the reader to state that models provided hourly J-values for a single day of August for the domains measured in ATom-1.

*Done. The model data provided here and archived under the doi is one-day global (for 24 hours). Sorry for the confusion.*

4). L30: what is the statistical picture of the impact of clouds on J-values established by the ATom-1 measurements?
5). L31-32: "the models show largely disparate patterns", "there is some limited, broad agreement": what are the disparate patterns and broad agreement, specifically?

*This section has been revised to give more information, but there remains no single zinging result.*

P4, L18: in Table 1

*Done, thanks.*

P5, L14-16: "It was not possible to have all the models simulate the flight paths and times" - why? Do most of these models use either assimilated or nudged meteorology? "we are trying to develop a climatology" - do you mean "statistics"? "the models were asked to pick a single day in August as representative of the cloud statistics over the large geographic blocks" - A single day of August 1-31, 2016, or near the end of the deployment (prior to Aug.6)? Figure S6 caption mentioned "one day in mid-August". Indicating the selected dates in the "Cloud data" column of Table 1 would help the readers who may want to reproduce or compare the results.

*See the Big-Topics overview above. All the models could not do the same year for this project. We now include the Year-Month-Day in Table 1 as requested.*

P6, 2nd paragraph: It's worth mentioning and explaining why the GC and GMI models, both driven by the MERRA-2 reanalysis, show large differences in JO1D over the tropical Pacific (upper left panels, Figs. 1-2).

*This is now noted in the paper, but the cause is not clear: tropical ozone columns are similar; perhaps different Fast-J datasets?*

P9, L11: the sharp peaks P9, L25: superscript "3/2"

*Yes, done.*

P13, L9, L14: "an observed climatology", "this CAFS climatology" - see above about "climatology".

*Yes, this has been cleaned up, see overview.*

P20, Table 1: GC "Cloud data" - the model uses liquid and ice cloud optical depths (not liquid and ice water) taken from MERRA-2. GC "Model references" - Gelaro et al. (2017) is the reference for MERRA-2. Cite Liu et al. (JGR 2006, 2009) for GC J-values and B-averaging.

*Done.*

P20, Table 1: 4th column, GFDL and GMI - what does C1 mean here? GMI "Cloud data" - the model uses liquid and ice cloud optical depths (not liquid and ice water) taken from MERRA-2.

*C1 refers to the cloud droplet size distribution (Deirmendjian, 1969) as used in Fast-J. And the table now notes this.*

Fig. S1 caption: low SZAs, not high SZAs Fig. S2 caption: "in the in the"
*Yes, done, thanks.*

Fig. S6 caption: Averaged in-cell cloud optical depth (COD, at 500-600 nm and .....); "blocks studies here blocks".
*Yes, fixed. Thanks.*

W.J. Collins (Referee)

Review of Hall et al. 2018
This paper nicely describes J-value measurements as part of the Atom-1 campaign. These measurements and those from further Atom campaigns will provide an excellent resource for assessing chemistry models.

*Thanks.*

The overall conclusion seems to be that "A primary uncertainty remains in the role of clouds in chemistry. . .". There should be enough information available to the authors to make some more conclusive statements. The authors need to think more about what are the key messages they want people to take away from reading this paper.

*Correct.  See overview discussion.  We have strengthened the last section and made the abstract clearer with some of the results.  The problem remains that the comparison we seek is difficult, our results include some clear successes and other places where more work is needed.  For example, the observed cloud-reduction in J-values in the uppermost troposphere are not reproduced by the physics of any of the models.  We need to look carefully at those data and the flight conditions under which they were made, as well the TUV modeling of the clear-sky conditions.  This shift in rlnJ is -0.05, or about 5%.*

The paper does not appear very clear as to its main purpose. Model-measurement comparisons can be used to assess whether models are fit for purpose (which doesn't seem the aim here) or to investigate where the models can be improved. This paper touches on the latter, suggesting that there are deficiencies in the treatment of clouds, but not what. It could be that models have too much or too little cloud, or that the frequency pdf of cloud amount is wrong, or that the radiation codes don't treat overlaps correctly or the parameterisation of cloud scattering is wrong. From this paper we still have no idea where to start looking to improve the models. Given the very different results between the observations and (some of) the models it must be possible to give some more concrete statements.

*See discussion above.  We have taken on the whole picture of the specific questions you list here because the real world consolidates all these.  There are many 'degrees of freedom' in these calculations and most of us have done many PhotoComp-like comparisons to address individual model differences.  This is the first attempt to diagnose these differences under realistic, climatologically varying conditions.  Even the cloud conditions have large uncertainty and cannot simply be fixed for the models from observations.  We could declare a full set of cloud data (cloud fraction on a grid cell, ice/liquid effective radius and water path) from some model, but implementing that data in the other 3D models would be painstaking, ranging from using the physics to get optical depth to simply re-gridding. Nevertheless we have added text to clarify the discrepancies between the models and data to elucidate areas for additional study.*

A very serious deficiency of the paper is the lack of coincident cloud observations. Cloudy/clear ratios are presented but with no data on what amounts of cloud caused these, it is therefore impossible to know whether the model-measurement differences are caused purely by different clouds. Measurements over 4 days in one particular August cannot be considered a representative

climatology. Were these 4 days more or less cloudy than the climatological average? Where would the observed clouds for these 4 days lie on figure S6? Even if clouds weren't specifically measured by the Atom-1, cloud data will be available from satellite measurements or re-analysis data. This will require a significant amount of work by the authors, but I don't see the value of the paper without this.

> *The authors respectfully disagree and refer to the discussion above regarding cloud observations. Coincident cloud observations would require 3D lidars, and then modeling would have to use 3D radiative transfer. The authors think this work has value and is the essential first step in evaluating photolysis and photochemistry in a realistic atmosphere. Adding a cloud-field analysis is beyond the scope of this paper but would be an important next step*

A significant improvement to the paper would come from plotting the cloudy/clear ratios as a function of cloud fraction (at different levels) for both the models and observations (or reanalysis) – or the authors may come up with a better way of controlling for cloud amount. This would overcome a lot of the issue of likely very different clouds amounts in the 4 particular observed August days with 1 random modelled August day. It would identify whether for the same cloud amount, models and measurements were calculating very different J-values.

> *See above. Not sure what you mean by 'cloud amount'; every model has a different definition of that. We did not ask the contributors to archive all of their cloud data because the data would not be commensurate. Also it might have caused some participants to drop out, thus limiting the comparison.*

Abstract: This needs to contain the key messages from the study.

> *Agreed. Done.*

Page 1: This page seems mostly trying to justify why it is not sensible to do point comparisons. While the real world is 3D, the model radiation codes are typically 1D and so comparing the 1D models with 3D observations would actually seem a sensible test of whether 1D models can represent the real world. More importantly, the claim that 4 days of measurements can be considered a climatology representative even of August 2016 let alone Augusts in general is never questioned. No evidence is presented as to how representative the observed statistics of clear/cloudy might be.

> *See discussion above. We agree and have withdrawn the 'representative climatology' statements and have added a test of model 'representativeness' with a new figure and discussion of sub-sampling of the Pacific block. Clearly we cannot state how representative the ATom-1 data are at this time. However, we will have the opportunity after adding the data from the last 3 deployments, which should be available early next year.*

Page 2, line 8: ". . . net probabilistic distribution of observed J-values". Again this glosses over the issue that the pdfs might be specific to those particular 4 days in August 2016, rather than being a more general climatological probabilistic distribution.

> *Page 3. Yes, just use 'statistics' – it is really the same thing.*

Page 5, line 16: Just as 4 days of observations can't be considered a climatology, a single day in August can't be representative. It should be possible to get much more than a single day's data from these models. Most of the models seem either to use re-analysis or were nudged; did they use meteorological fields from the days of the campaign or a random August?

*The models used various meteorological data that were noted in Table 1. The years are varied. Not everyone was able to use 2016 when we collected the data and we did not want to delay the analysis. Now we have added the year-month-day information to that table.*

Page 7, line 19: Are the narrow peaks due to the frequency of clouds? It could be that some models have an average cloud fraction of 0.1 by having a fraction of 0.1 every- where, but others have the same mean fraction with a large variety of cloud amounts. PDFs of CF and/or COD are needed (along the lines of fig 6).

*The narrow peaks mean simply that for 40% of the time or more, these models have J-values that are within ±2.5% of clear sky – see Figure 5. The cause could be manifold: very thin OD, method of cloud overlap, and even coding errors. CF does not matter unless COD is large enough. For the B-averaging codes, it is $CF^{3/2}xOD$. Comparing details of the cloud data is essential for a single fixed-case photocomp study, but would not clarify the model differences here.*

Section 3.2: Given the large differences between the models it is essential to use more than the averaged cloud properties to understand the correlations between CF, COD and J-values, and how these differ between the models. How is COD calculated – does it use the same radiation scheme as for J-values or is from the climate radiation scheme?

*See discussion above. The correlations would require an immense new dataset and still not answer some of these questions. A model-model study on clouds and how to implement them would be best addressed by the CCM community.*

Section 4.3: It would be useful to use UCI to replicate figure 4 for the ICAs and QCAs to determine how much the averaging contributes to the different distributions in observations and models.

*Not sure what this comment is about. UCI standard model runs with QCAs generating a single J-value and a second model version with Briegleb averaging is included (UCIb). To extract the J-values from the individual ICAs is prohibitive since it would increase the cost of the UCI simulation 100x or more. Further, it would require a code re-write to extract sub-resolved J's (ICAs or QCAs) from within the parallelization. The Cloud-J paper (2015) did this with a single fixed-case study.*

Conclusions: This paper needs a conclusion section. As it stands it is not at all obvious what the overall conclusion of the study is, other than clouds are difficult and more work is needed.

*Conclusions have been updated and are more focused, but an important conclusion remains that which is reflected in your last clause. This paper provides a new statistical analysis to study the effect of clouds on photolysis and hence photochemistry. The limitations of this study are noted, yet the analysis already highlights new aspects (or problems) of the models and differences in modeling techniques.*

---

## Author Response (AR1)

**Discussion of Big-picture Topics**, followed by the individual reviews and our responses.

*"climatology"*
This appears to be a loaded word, and we have used it too carelessly. We have shifted a lot of "climatology(ies)" to "statistics" as recommended, but some use of the word is retained since it is the intent to build such a climatology. Thus the title has been changed to "building a climatology." The models are tested as to the robustness of their statistics in terms of space, but we did not go back and ask for full resubmissions with different years. We have previously shown that sampling along a single meridian in the tropical Pacific produces a robust 2D distribution of chemical species (see Fig 8 below from the Prather et al: Atmos. Chem. Phys., 17, 9081–9102, 2017, https://doi.org/10.5194/acp-17-9081-2017), and have added a simple test here by splitting the tropical Pacific block into east and west halves.

*ATom-1 deployment only:*
One review was critical on the choice of dates and wished to put the "first deployment" with dates in the title. We tried several versions and it was just too clumsy, and not really title material. Thus we have brought forward in the abstract and paper the fact that we are only presenting ATom-1 data. The dates are now specific: the ATom-1 deployment consisting of 10 research flights was 29 Jul to 23 Aug 2016. While the full ATom-1 measurements of J-cloudy and J-clear are reported and archived with this paper, we analyze here only the Pacific data north of 20ºS, which includes research flights 1-5 (29-Jul to 8-Aug). Likewise, the current model analysis includes only the two Pacific blocks from 20ºS to 50ºN, but the archived model data is global and for a mid-August day (varying years). Thus, later analyses can look at the Southern or Atlantic Oceans (or continental data) using the archived data sets.

*Sampling of different years, more extensive cloud statistics, repeatability of just one day in August:*
We agree with reviewer Collins that it would be wonderful to have done this analysis for the full month of August and for different years to see if the modeled J-values statistics were robust and did not shift. Given the scope of the current work, and the effort from the modeling community, that would not have been possible, and we would have lost participants. As it was, and including some mistakes in diagnostics, the models often had to be corrected and re-run, and the amount of data processed was >60 GB.

The reason for selecting large geographic blocks and 24 hours of model data were to make more robust statistics. Given the limits on available data, we have gone back to the J-values here and sub-sampled our declared "tropical Pacific" region into a west and east half, and also into a narrow stripe down the dateline. This was a useful exercise and the new results are presented for all models in Supplementary figures and discussed in the main text. Basically, most models show no difference across the sub-sampling (NCAR, GFDL, IFS, UCI, UKCA, MOCA) while three models (GC, GMI, GISS) show weaker cloud effects in the East half of the region. This is consistent in that these three use MERRA cloud fields in one way or another.

There was some wish for further analysis of the cloud fields (which would require ice and liquid water paths, effective radii and cloud fraction) and their correlation with J-value anomalies. This new study would be interesting but require much more extensive work from the modelers, and of course, more detailed comparisons of how the models implement the cloud data. This would be better as a model-model set of fixed case studies.

*Why not just analyze the cloud fields?  or use assimilated/nudged cloud fields?*
We chose not to focus on comparing the modeled and observed cloud fields because they really do not tell us how J-values are perturbed by clouds.  Moreover, for the observations, there are no observed cloud fields at the resolution of the airplane measurements. J-values vary at the 3-sec resolution of the data (0.8 km) and 3D cloud fields at this scale are not available for the flight from either the aircraft or satellite.  There are some marvelous 3D high-resolution cloud fields (on the scale of the aircraft variability), but these are limited to CloudSat-CALIPSO overpasses that have been extended to 200-km wide swaths (see Barker et al., 2011; Miller et al., 2014), not to the 2,000+ km swaths needed to get daily coverage.   Even once-a-day coverage will not define clouds over the ATom flight.  Assimilating clouds is still a difficult, unsolved problem (from Alan Geer, ECMWF, 2017):

- forecast models still use simplified representations of cloud and convection
- the radiative transfer models required to simulate such 3D structures accurately would be far too expensive for operational use, and
- the predictability of cloud and precipitation tails off in a matter of hours.

A 2018 effort to assimilate clouds in high-resolution models (A.T. White. "Improving Cloud Simulation for Air Quality Studies through Assimilation of Geostationary Satellite Observations in Retrospective Meteorological Modeling", MWR, https://doi.org/10.1175/MWR-D-17-0139.1) is still based on the model physics that creates clouds: "The basic approach is to create positive vertical motion within the model to produce clouds and negative vertical motion to dissipate clouds, based on GOES cloud fields." So we cannot simply use the assimilated or 'nudged' cloud fields.  (Nudging uses U, V, T, q not clouds).

An additional reason why we do not spend more time on analyzing the cloud data from the models is that our primary objective is to find out how photochemistry, through J-values, is affected by clouds.  The way that models implement clouds can be very different, even when given the same key data: water path and effective radius for ice and liquid, and cloud fraction.  In the example pointed out by the reviewers, GC and GMI are both using MERRA clouds and should have similar results:  yet, they differ in clear-sky J's; but this difference become greater when we look at all-sky (cloudy) J's.  Our analysis here is not a thorough single-test-case model comparison in which one might be able to identify why the models take similar cloud data and use almost the same Fast-J code to give different cloud effects.  That would be interesting, but is beyond the scope of this paper.

Barker, H. W., M. P. Jerg, T. Wehr, S. Kato, D. P. Donovan, and R. J. Hogan, 2011: A 3D cloud-construction algorithm for the EarthCARE satellite mission. Quart. J. Roy. Meteor. Soc., 137, 1042–1058, doi:10.1002/qj.824.
Miller, S.D., J.M. Forsythe, P.T. Partain, J.M. Haynes, R.L. Bankert, M. Sengupta, C. Mitrescu, J.D. Hawkins, and T.H. Vonder Haar (2014) Estimating three-dimensional cloud structure via statistically blended satellite observations. J. Appl. Meteor. Climatol., 53, 437–455, doi:10.1175/JAMC-D-13-070.1

**Summary.**  Given that this paper is a start at analyzing how clouds control the statistical variation of J-values, it cannot answer all the questions we have.   The reviewers have come up with an interesting set of follow-on questions, which we would all like to see answered.  We believe that that we have presented a thorough, innovative approach for assessing how models implement clouds in their photochemistry, and come up with a new type of observational dataset to test this.  Some large questions remain as noted in the summary discussion (Section 5).  We believe the current content is extensive but necessary to understand this work.  Let this be the first step.

[Figure]

**Figure 8.** AIR-weighted 2-D probability distributions for $NO_x$ vs. HOOH averaged over tropical Pacific block (150–210° E, 20° S–20° N, 0–12 km) and for different single-longitude transects from 150–210° E, shown for models **(a)** C (GFDL-AM3) and **(b)** F (UCI-CTM). The fitted 2-D ellipses are shown for the full block (thick black line) and five longitude transects (colored lines) for models **(c)** C and **(d)** F. The block ellipse for the other model is shown as a thin black dashed line.

Anonymous Referee #3

This paper describes the novel use of observation-based photolysis rates derived from aircraft profile flux measurements to test global atmospheric chemistry models. It adopts an original and innovative approach, and provides a highly valuable first step in the evaluation of cloud impacts on photolysis rates in models, deriving some useful pointers to areas that need further exploration or model development. The study is well executed and thorough, exploring sensitivity to ozone column and albedo as well as to a central focus on cloud properties. The analysis approaches appear sound, the conclusions are useful, and as such the paper merits publication in ACP once some minor issues have been addressed.

*Thank you.*

Specific comments
Data was provided from the models for a single day in August. How dependent are the model results on the specific day chosen? Are the regions large enough to provide a truly representative distribution of cloud coverage? This would be relatively simple to check.

*This is a tough question. Please see big-topic answers at the beginning of this response. In part answer your question, we redid the sampling in the tropical Pacific to test the statistics (new Figure S7).*

In addition, the number of observed samples per block is stated on page 5, but not the number of low-SZA model samples.

*Yes, done. Some numbers were added in text, others in the figure caption.*

The coarse resolution of the models leads to an averaging of cloud cover and an under-estimation of clear sky conditions (explored in section 4.3). What would the distribution of rlnJ look like for the observations if these were averaged to the physical scale of the models?

*Interesting question. Using data and analyses similar to that for Figure 6, we looked at what happened as we took cloud data from a 640x320 (1/2 degree) model and averaged it over320x160 and 160x80. There was a detectable shift to less-clear skies, but the much bigger effect was on how the cloud fractions are used (Fig. 6). Thus, we deemed it not critical for this paper.*

*The second part of your question appears to ask about averaging the ATom-1 J's to a 100-km grid. There is no natural registration for such a grid and we did not try it. This would be better when we have added the last 3 deployments and have a larger data set. This topic is very important for model development: the best we can hope for is that modeled J's on 10-100 km scales are effectively the average of the 3D high-resolution fields (i.e., the ATom J's). This is discussed in Section 5.*

To what extent does the noise inherent in the observation-based rlnJ values (evident in the broadening seen in Fig 4) wash out the vertical profile of cloud impacts on photolyis rates that is clearly seen in the models?

*A very good, but difficult question to answer at this stage. This paper is based on the first released J-cloudy/J-clear ATom-1 data, and it points to the need for further work on*

*tightening up the spread (noise) in the distribution that could be due to correctable physical processes such as ocean surface albedo, airplane maneuvers (banking and ascent/descent can change the orientation of the 2-pi detectors relative to the direct beam of the sun), etc. When all 4 deployments are available, we can revisit this along with developing an improved set of model diagnostics.*

Does the occasional enhancement of near-surface J-values and reduction at high altitudes reflect biases in model cloud distributions, 3-D effects, or just noise?

*The low level enhancements are likely due to thinnish clouds where the J's are enhanced within and below. The low-level enhancement in observed J-O1D Tr.Pac. is just not sensible in terms of the 1D modeling, and this is now noted. We need to develop statistics on 3D cloud effects at the scale seen by the aircraft to truly answer this question.*

Is Fig 2 based on additional model runs without clouds (alluded to on page 4, line 23) or on clear-sky columns that are a subset of the all-sky data shown in Fig 1? This should be stated on page 6. If the clear-sky values are a subset, what bias does the different locations and SZAs of clear and cloudy columns introduce?

*Sorry for the confusion, this is rewritten on page 4 to be explicit: The models submitted two full 24-hour simulations: one with their standard cloud treatment and one where the aerosols and clouds were zeroed. No subsetting.*

Fig 5 is an interesting and well thought out way of presenting the data, but is difficult for the reader to interpret, particularly where the bar indicating the proportion of reduced/enhanced J-values does not align with the cross indicating the mean magnitude.

*Yes, Fig 5 is a dense figure in terms of information, yet all that information needs to be put together to understand the J statistics. We have added an explanatory legend to Figure 5b (it has the smaller figure caption) and explained more in the text about the 5 values being plotted for each model and the CAFS data.*

Minor issues and Typos
Abstract, line 30: "more importantly" Please rephrase this or reorder the sentence appropriately.

*Yes, done. Abstract is substantially rewritten in response to reviews.*

Page 9, line 25: 3/2 should be a power

*Yes, done.*

Page 10, line 23: section 4.3 is important, but the title of the section is awkward, please consider rephrasing "finding clear sky".

*Yes, done. Thanks."4.3 Averaging over clouds affects clear-sky fraction"*

Page 11, line 6-7: this partly repeats information provided on page 5.

*Yes, had dropped references and shortened a bit on page 11.*

The caption to Figure 6 is too long. The information is valuable, but some of it should be included in the text on page 11. (Typo: please replace star with multiplication sign for consistency)

*Agreed. Cut caption, added some discussion to text.*

Caption to Fig S2, one line from bottom: "in the in the"
*Yes, done.*

Caption to Fig S6, line 2: remove second occurrence of "blocks"
Fig S6 needs to be cleaned up so that the titling is legible and the model legend is not superimposed on the axis labels.
*Yes, both done.*

Anonymous Referee #2

This paper presented a statistical analysis of the photolysis rates data obtained over the Pacific during the first deployment of the ATom mission (July 29-Aug.6, 2016) and evaluated the performance of nine global models with respect to how clouds affect photolysis rates (J-values). The ATom J-values are a unique data set for testing how well clouds are treated in terms of their impact on photochemistry in current global models. The model J-values presented in this study are from simulations with each model's own configuration without controlling input conditions among all nine models. Comparisons of the probability distributions of cloud impacts on J-values between the models and measurements provided original insights into model performance differences. The paper concludes with an interesting discussion on the difficulties and challenges involved in simulating cloud impacts on J-values and comparing with measurements. The need for more effort to characterize the main factors contributing to the model differences are also acknowledged and discussed. The paper is overall well written and I recommend publication on ACP after some minor modifications.

> *Thank you.*

Title: The title would suggest this paper is about the analysis of photolysis rates from all ATom campaigns, but in fact only the data from ATom-1 (July 29-Aug.6, 2016) is used to test nine global models. The title should reflect these.
Also "climatology" (a term used throughout the text) cannot be derived from a single deployment. "Statistics" is probably a more appropriate term as part of the title for the kind of analysis presented, and is often used in the text.

> *See overview discussion on these 'big topics'. We now use 'statistics' more and 'climatology' less, but we still have the objective of developing a climatology for all models. Thus the title was changed to indicate the intent to build a climatology. It was impossible to put in all the information about the deployment in the title, and so we put it in the abstract and the opening sections.*
>
> *Also the published data set (full ATom-1) and the analysis here are clearly stated in the abstract and intro.: "This analysis is limited to the first half of the deployment over the Pacific Ocean, but publishes the complete data set."*

Abstract: The abstract could be improved: 1). it doesn't flow that well.
> *A substantial rewrite was done. Yes, it needed one.*

2). P1, L28: "during the first deployment (ATom-1) in August 2016": it's actually July 29-August 6, 2016.
> *Yes, corrected. The ATom-1 deployment (in air) was 29 Jul to 23 Aug. The data from the tropical Pacific was taken between 29 Jul and 8 Aug (sorry for the 6 Aug typo in the manuscript). This is now accurately described in the paper. The model data is global and the ATom-1 CAFS data archived with this paper are complete, all 10 research flights, 29 Jul – 23 Aug 2016.*

3). P1, L29: it would be useful to the reader to state that models provided hourly J-values for a single day of August for the domains measured in ATom-1.

*Done. The model data provided here and archived under the doi is one-day global (for 24 hours). Sorry for the confusion.*

4). L30: what is the statistical picture of the impact of clouds on J-values established by the ATom-1 measurements?

5). L31-32: "the models show largely disparate patterns", "there is some limited, broad agreement": what are the disparate patterns and broad agreement, specifically?

*This section has been revised to give more information, but there remains no single zinging result.*

P4, L18: in Table 1

*Done, thanks.*

P5, L14-16: "It was not possible to have all the models simulate the flight paths and times" - why? Do most of these models use either assimilated or nudged meteorology? "we are trying to develop a climatology" - do you mean "statistics"? "the models were asked to pick a single day in August as representative of the cloud statistics over the large geographic blocks" - A single day of August 1-31, 2016, or near the end of the deployment (prior to Aug.6)? Figure S6 caption mentioned "one day in mid-August". Indicating the selected dates in the "Cloud data" column of Table 1 would help the readers who may want to reproduce or compare the results.

*See the Big-Topics overview above. All the models could not do the same year for this project. We now include the Year-Month-Day in Table 1 as requested.*

P6, 2nd paragraph: It's worth mentioning and explaining why the GC and GMI models, both driven by the MERRA-2 reanalysis, show large differences in JO1D over the tropical Pacific (upper left panels, Figs. 1-2).

*This is now noted in the paper, but the cause is not clear: tropical ozone columns are similar; perhaps different Fast-J datasets?*

P9, L11: the sharp peaks P9, L25: superscript "3/2"

*Yes, done.*

P13, L9, L14: "an observed climatology", "this CAFS climatology" - see above about "climatology".

*Yes, this has been cleaned up, see overview.*

P20, Table 1: GC "Cloud data" - the model uses liquid and ice cloud optical depths (not liquid and ice water) taken from MERRA-2. GC "Model references" - Gelaro et al. (2017) is the reference for MERRA-2. Cite Liu et al. (JGR 2006, 2009) for GC J-values and B-averaging.

*Done.*

P20, Table 1: 4th column, GFDL and GMI - what does C1 mean here? GMI "Cloud data" - the model uses liquid and ice cloud optical depths (not liquid and ice water) taken from MERRA-2.

*C1 refers to the cloud droplet size distribution (Deirmendjian, 1969) as used in Fast-J. And the table now notes this.*

Fig. S1 caption: low SZAs, not high SZAs Fig. S2 caption: "in the in the"
*Yes, done, thanks.*

Fig. S6 caption: Averaged in-cell cloud optical depth (COD, at 500-600 nm and .....); "blocks studies here blocks".
*Yes, fixed. Thanks.*

W.J. Collins (Referee)

Review of Hall et al. 2018
This paper nicely describes J-value measurements as part of the Atom-1 campaign. These measurements and those from further Atom campaigns will provide an excellent resource for assessing chemistry models.

*Thanks.*

The overall conclusion seems to be that "A primary uncertainty remains in the role of clouds in chemistry. . .". There should be enough information available to the authors to make some more conclusive statements. The authors need to think more about what are the key messages they want people to take away from reading this paper.

*Correct. See overview discussion. We have strengthened the last section and made the abstract clearer with some of the results. The problem remains that the comparison we seek is difficult, our results include some clear successes and other places where more work is needed. For example, the observed cloud-reduction in J-values in the uppermost troposphere are not reproduced by the physics of any of the models. We need to look carefully at those data and the flight conditions under which they were made, as well the TUV modeling of the clear-sky conditions. This shift in rlnJ is -0.05, or about 5%.*

The paper does not appear very clear as to its main purpose. Model-measurement comparisons can be used to assess whether models are fit for purpose (which doesn't seem the aim here) or to investigate where the models can be improved. This paper touches on the latter, suggesting that there are deficiencies in the treatment of clouds, but not what. It could be that models have too much or too little cloud, or that the frequency pdf of cloud amount is wrong, or that the radiation codes don't treat overlaps correctly or the parameterisation of cloud scattering is wrong. From this paper we still have no idea where to start looking to improve the models. Given the very different results between the observations and (some of) the models it must be possible to give some more concrete statements.

*See discussion above. We have taken on the whole picture of the specific questions you list here because the real world consolidates all these. There are many 'degrees of freedom' in these calculations and most of us have done many PhotoComp-like comparisons to address individual model differences. This is the first attempt to diagnose these differences under realistic, climatologically varying conditions. Even the cloud conditions have large uncertainty and cannot simply be fixed for the models from observations. We could declare a full set of cloud data (cloud fraction on a grid cell, ice/liquid effective radius and water path) from some model, but implementing that data in the other 3D models would be painstaking, ranging from using the physics to get optical depth to simply re-gridding. Nevertheless we have added text to clarify the discrepancies between the models and data to elucidate areas for additional study.*

A very serious deficiency of the paper is the lack of coincident cloud observations. Cloudy/clear ratios are presented but with no data on what amounts of cloud caused these, it is therefore impossible to know whether the model-measurement differences are caused purely by different clouds. Measurements over 4 days in one particular August cannot be considered a representative

climatology. Were these 4 days more or less cloudy than the climatological average? Where would the observed clouds for these 4 days lie on figure S6? Even if clouds weren't specifically measured by the Atom-1, cloud data will be available from satellite measurements or re-analysis data. This will require a significant amount of work by the authors, but I don't see the value of the paper without this.

*The authors respectfully disagree and refer to the discussion above regarding cloud observations. Coincident cloud observations would require 3D lidars, and then modeling would have to use 3D radiative transfer. The authors think this work has value and is the essential first step in evaluating photolysis and photochemistry in a realistic atmosphere. Adding a cloud-field analysis is beyond the scope of this paper but would be an important next step*

A significant improvement to the paper would come from plotting the cloudy/clear ratios as a function of cloud fraction (at different levels) for both the models and observations (or reanalysis) – or the authors may come up with a better way of controlling for cloud amount. This would overcome a lot of the issue of likely very different clouds amounts in the 4 particular observed August days with 1 random modelled August day. It would identify whether for the same cloud amount, models and measurements were calculating very different J-values.

*See above. Not sure what you mean by 'cloud amount'; every model has a different definition of that. We did not ask the contributors to archive all of their cloud data because the data would not be commensurate. Also it might have caused some participants to drop out, thus limiting the comparison.*

Abstract: This needs to contain the key messages from the study.

*Agreed. Done.*

Page 1: This page seems mostly trying to justify why it is not sensible to do point comparisons. While the real world is 3D, the model radiation codes are typically 1D and so comparing the 1D models with 3D observations would actually seem a sensible test of whether 1D models can represent the real world. More importantly, the claim that 4 days of measurements can be considered a climatology representative even of August 2016 let alone Augusts in general is never questioned. No evidence is presented as to how representative the observed statistics of clear/cloudy might be.

*See discussion above. We agree and have withdrawn the 'representative climatology' statements and have added a test of model 'representativeness' with a new figure and discussion of sub-sampling of the Pacific block. Clearly we cannot state how representative the ATom-1 data are at this time. However, we will have the opportunity after adding the data from the last 3 deployments, which should be available early next year.*

Page 2, line 8: ". . . net probabilistic distribution of observed J-values". Again this glosses over the issue that the pdfs might be specific to those particular 4 days in August 2016, rather than being a more general climatological probabilistic distribution.

*Page 3. Yes, just use 'statistics' – it is really the same thing.*

Page 5, line 16: Just as 4 days of observations can't be considered a climatology, a single day in August can't be representative. It should be possible to get much more than a single day's data from these models. Most of the models seem either to use re-analysis or were nudged; did they use meteorological fields from the days of the campaign or a random August?

*The models used various meteorological data that were noted in Table 1. The years are varied. Not everyone was able to use 2016 when we collected the data and we did not want to delay the analysis. Now we have added the year-month-day information to that table.*

Page 7, line 19: Are the narrow peaks due to the frequency of clouds? It could be that some models have an average cloud fraction of 0.1 by having a fraction of 0.1 every- where, but others have the same mean fraction with a large variety of cloud amounts. PDFs of CF and/or COD are needed (along the lines of fig 6).

*The narrow peaks mean simply that for 40% of the time or more, these models have J-values that are within ±2.5% of clear sky – see Figure 5. The cause could be manifold: very thin OD, method of cloud overlap, and even coding errors. CF does not matter unless COD is large enough. For the B-averaging codes, it is $CF^{3/2}xOD$. Comparing details of the cloud data is essential for a single fixed-case photocomp study, but would not clarify the model differences here.*

Section 3.2: Given the large differences between the models it is essential to use more than the averaged cloud properties to understand the correlations between CF, COD and J-values, and how these differ between the models. How is COD calculated – does it use the same radiation scheme as for J-values or is from the climate radiation scheme?

*See discussion above. The correlations would require an immense new dataset and still not answer some of these questions. A model-model study on clouds and how to implement them would be best addressed by the CCM community.*

Section 4.3: It would be useful to use UCI to replicate figure 4 for the ICAs and QCAs to determine how much the averaging contributes to the different distributions in observations and models.

*Not sure what this comment is about. UCI standard model runs with QCAs generating a single J-value and a second model version with Briegleb averaging is included (UCIb). To extract the J-values from the individual ICAs is prohibitive since it would increase the cost of the UCI simulation 100x or more. Further, it would require a code re-write to extract sub-resolved J's (ICAs or QCAs) from within the parallelization. The Cloud-J paper (2015) did this with a single fixed-case study.*

Conclusions: This paper needs a conclusion section. As it stands it is not at all obvious what the overall conclusion of the study is, other than clouds are difficult and more work is needed.

*Conclusions have been updated and are more focused, but an important conclusion remains that which is reflected in your last clause. This paper provides a new statistical analysis to study the effect of clouds on photolysis and hence photochemistry. The limitations of this study are noted, yet the analysis already highlights new aspects (or problems) of the models and differences in modeling techniques.*

[revised manuscript text omitted]
 2). For UCI the number of points is 240,000 (block 1) and 120,000 (block 2); while for the higher resolution NCAR model, the values are 1,400,000 and 700,000, respectively.

[Figure]

Figure S2. Mean CAFs profiles of J-O1D (left) and J-NO2 (right) from ATom-1 made under all-sky conditions for high-sun conditions, cos(sza) > 0.8. The 3 geographic blocks shown here are the standard blocks 1 (blue, Tropical Pacific, 20 ºS – 20 ºN x 160 ºE – 240 ºE), and 2 (red, North Pacific, 20 ºN – 50 ºN x 170 ºE – 225 ºE, plus a global block (black, 55 ºS – 55 ºN, all longitudes, including the Atlantic). Each 3-sec observation is averaged with equal weight in 100-hPa pressure bins. The standard deviation for block 1 (blue, *) is shown along with that from the corresponding TUV model calculation for clear sky (blue, o). The patterns here are expected: J-O1D is sensitive to $O_3$ absorption and hence has higher upper-tropospheric values in the tropics; J-NO2 is more sensitive to cloud scattering and hence has higher values in the  lower troposphere of the North Pacific. The standard deviations show that J-O1D variance is driven by $O_3$ and SZA, and not by clouds; while the J-NO2 shows the opposite.

[Figure]

Figure S3. Total ozone columns (in DU) versus latitude in August for 9 models and 8 years of OMI observations (2010-2017, Levelt et al., 2006; Veefkind et al., 2006). Most curves are the monthly zonal means, but some data are from a single day, or just the ATom flight track for several days (e.g., GFDL, explains the lack of smoothness). Differences here can explain only some of the spread in clear-sky J-O1D values in Figure 2.

[Figure]

Figure S4. Effects of clear sky (blue points), simple averaged clouds (gray points), and fractional cloud overlap (the reference case, 1:1 dashed line) using the UCI CTM. X-axis is the same as the Y-axis, but for the reference model UCI 2016. Direct parcel-by-parcel comparison of modeled 24-hour reactivities (top row: P-O3, L-O3, L-CH4; all ppb/day) and photolysis rates (J-NO2, J-O1D; all /sec) calculated for a data stream of 14,880 simulated air parcels from 60 °S to 60 °N, 0.5 km to 12 km altitude, and along 180 °W, see Prather et al., 2018. Each point is an average of the 5 simulated dates in August (8/01, 8/06, 8/11, 8/16, 8/21). UCI 2016 ref uses full cloud quadrature and the newly implemented decorrelation lengths for cloud overlap (Prather, 2015). From the figure, the different cloud treatments are clearly visible in the shifts in J-NO2 and J-O1D, and they have largest effect on P-O3 as compared with L-O3 and L-CH4. On average, the clear-sky simulation has 3-4% lesser J-values and similar changes in all 3 reactivities. The averaged-cloud method has about 12 % greater J-values (mostly above clouds), increasing reactivities by 5 % (L-O3 and L-CH4) to 10% (P-O3).

[Figure]

Figure S5. Histogram of the natural log of the ratio of cloudy-to-clear J values (J-O1D & J-NO2), designated rlnJ, which is calculated from the 10-sec CAFS data and sorted into 3 pressure bins (blue, 100 – 300; red, 300 – 900; black, 900 – surface hPa). Note that the upper level has a peak near 0.0, corresponding to clear sky; the lowest level extends more to values <0; and that the N. Pacific for <900 hPa has a large number of cloud enhanced J-values (rlnJ >0). The CAFS data were binned at 0.01 rlnJ and smoothed with a 1-2-1 filter six times.

[Figure]

Figure S6.  Averaged in-cell cloud optical depth (COD, at ~600 nm and per 100 hPa) and cloud fraction (CF) over the two geographic blocks studied here blocks (Tropical Pacific, 20 ºS – 20 ºN x 160 ºE – 240 ºE, and North Pacific, 20 ºN – 50 ºN x 170 ºE – 225 ºE).  Averages are made over 24 hours for one day in mid-August as used in this paper.  For these data (unlike the J-value statistics), all hours are weighted equally independent of solar zenith angle. Column total COD for each region is given in the order/color-indexed numbers on the right of each COD panel.

[Figure]

Figure S7a.  The effect of sampling different regions of the tropical Pacific in terms of the frequency of occurrence and magnitude of changes caused by clouds in J-NO2 at 100-300 hPa. See Figure 5.  Each of the 9 panels shows a single model with different sampling and also the CAFS observations over the tropical Pacific (bottom black bar and X's).  The top purple bar is the entire tropical Pacific as defined in Figure 5 (20S-20N, 160E-240E) and the next three bars (blue, green, yellow) sub-sample this region into West (160E-200E), East (200E-240E) and 180E (175E-185E), respectively.  The next two bars are the North Pacific (as in Figure 5) and a global sampling (50S-50N, all longitudes).

[Figure]

Figure S7b. The effect of sampling different regions of the tropical Pacific in terms of the frequency of occurrence and magnitude of changes caused by clouds in J-NO2 at surface-900 hPa. See Figure S7Xa.

[Figure]

Figure S8. The ratio of J-cloudy/J-clear calculated for J-O1D (left) and J-NO2 (right) at 300 hPa for a marine stratus cloud (CF = 100%) at about 900 hPa. The natural log of the ratio (rlnJ) is calculated for a cloud optical depth (COD) ranging from 0.1 to 100. The calculation uses Cloud-J (Prather, 2015), which does not scale COD, uses the truncated (order 8) phase function, and applies a constant lower boundary albedo = 0.05. The incident SZA ranges from 0º (black) to 60º (red) in 10º intervals (blue). The rlnJ's in this paper are restricted to 0º – 40º. A 5 % enhancement (rlnJ = 0.05) occurs at COD = 2 for J-O1D and COD = 1 for J-NO2, demonstrating the greater sensitivity of J-NO2 to clouds.

[Figure]

Figure S9. Probability distribution of the natural log of the ratio of J-values calculated using an interactive ocean surface albedo (OSA) to J-values using a fixed albedo. J-O1D values (left) and J-NO2 values (right) assume clear skies for both OSA and fixed-albedo. This format is similar to Figure 4. The interactive OSA model includes a dependence on wavelength and incident-angle now in Cloud-J v8, see text. The fixed albedo is denoted by 'a =' in the legend. These distributions show the relative error in J-values calculated in typical models using a fixed albedo instead of the more physically based OSA models (Séférian et al., 2018). For both cases the reflected sunlight is assumed to be isotropic. The OSA depends on surface wind (sampled uniformly from 1 to 21 m/s) and chlorophyll abundance (sampled uniformly in log from 0.01 to 30 mg/m$^3$). The plots are also split between the lowermost troposphere (lower panels, >850 hPa) and free troposphere (upper, 200 – 850 hPa).

| Table S1.  Model contact information | | | |
|---|---|---|---|
| short name | long name | POCs for these simulations | |
| CAFS | --- | Sam Hall <halls@ucar.edu> | Kirk Ullmann  |
| GC | GEOSChem | Murray, Lee <lee.murray@rochester.edu> | |
| GFDL | GFDL AM3 | Arlene Fiore <amfiore@ldeo.columbia.edu> | Gustavo Correa <gus@ldeo.columbia.edu> |
| GISS | GISS ModelE2 | Murray, Lee <lee.murray@rochester.edu> | |
| GMI | GSFC GMI | Sarah A. Strode <Sarah.A.Strode@nasa.gov> | Stephen Steenrod <Stephen.D.Steenrod@nasa.gov> |
| IFS | ECMWF IFS | Johannes Flemming <johannes.flemming@ecmwf.int> | Vincent Huijnen <vincent.huijnen@knmi.nl> |
| MOCA | MOCAGE | Beatrice Josse  | Jonathan Guth <jonathan.guth@meteo.fr> |
| NCAR | CESM | Jean-Francois Lamarque <lamar@ucar.edu> | |
| UCI | UCI CTM | Michael Prather <mprather@uci.edu> | Clare Flynn <claref@uci.edu> |
| UKCA | UKCA | Luke Aabraham <luke.abraham@atm.ch.cam.ac.uk> | Alex Archibald <ata27@cam.ac.uk>, Marcus Koehler <m.koehler@uea.ac.uk> |